# Regularized Latent Dynamics Prediction is a Strong Baseline for Behavioral Foundation Models

**Pranaya Jajoo**[1,2*]    **Harshit Sikchi**[4,†]    **Siddhant Agarwal**[4,†]

**Amy Zhang**[4]    **Scott Niekum**[5]    **Martha White**[1,2,3]

[1] Department of Computing Science, University of Alberta, Canada
[2] Alberta Machine Intelligence Institute (Amii)
[3] Canada CIFAR AI Chair
[4] The University of Texas at Austin
[5] University of Massachusetts Amherst
[†] Equal contribution

## Abstract

Behavioral Foundation Models (BFMs) produce agents with the capability to adapt to any unknown reward or task. These methods, however, are only able to produce near-optimal policies for the reward functions that are in the span of some pre-existing *state features*, making the choice of state features crucial to the expressivity of the BFM. As a result, BFMs are trained using a variety of complex objectives and require sufficient dataset coverage, to train task useful spanning features. In this work, we examine the question: are these complex representation learning objectives necessary for zero-shot RL? Specifically, we revisit the objective of self-supervised next-state prediction in latent space for state feature learning, but observe that such an objective alone is prone to increasing state-feature similarity, and subsequently reducing span. We propose an approach, Regularized Latent Dynamics Prediction (`RLDP`), that adds a simple orthogonality regularization to maintain feature diversity and can match or surpass state-of-the-art complex representation learning methods for zero-shot RL. Furthermore, we empirically show that prior approaches perform poorly in low-coverage scenarios where `RLDP` still succeeds.

## 1 Introduction

Zero-shot reinforcement learning (RL) (Touati et al., 2022) is a problem setting where we learn an agent that can solve *any* task in the environment without any additional training or planning, after an initial pretraining phase. Zero-shot RL has significant practical potential in developing generalist agents with wide applicability. For instance, robotics applications, like robotic manipulation or drone navigation, often require agents to solve a wide variety of unknown tasks. A general-purpose household robot needs to possess the capability to flexibly adapt to various household chores without explicit training for each new task.

Behavioral Foundation Models (BFMs) have been shown to be promising for zero-shot RL (Touati et al., 2022; Agarwal et al., 2024). BFMs are trained on a dataset of reward-free interactions, with the aim to provide a near-optimal policy for a wide class of reward functions without additional learning or training during test-time. BFMs are trained by a) learning a state representation $\varphi : s \to \mathbb{R}^d$, and b) learning a policy $\pi$ conditioned on a latent vector $z \in \mathbb{R}^d$, where the $z$ can be seen as a task embedding for reward $r(s) = \varphi(s)^\top z$. In this way, the BFM consists of a space of policies, where different policies can be extracted by querying the learned policy using different $z$. At test time, given any reward function $r^{test}(s)$, the near-optimal policy $\pi_{z_{r^{test}}}$ is obtained zero-shot by solving for $z_{r^{test}}$ such that $r^{test}(s) \approx \varphi(s)^\top z_{r^{test}}$. This assumption on the reward is also used for successor

---

*Correspondence to `pranayajajoo@ualberta.ca`

features (Barreto et al. (2016)), which consist of the discounted cumulative sum of feature vectors under a policy. Successor features zero-shot produce the action-values for a new reward, given by vector $z$, and BFM approaches often use successor features to learn the policies.

The performance of the BFM relies heavily on the state representation, which is both used to extract $z$ for the reward and for the policy. State-of-the-art methods Touati & Ollivier (2021); Agarwal et al. (2024); Park et al. (2024) usually learn state representations that retain information to represent *successor measures* under a wide class of policies. A successor measure captures the (discounted) state visitation of a policy, given any starting state. They are the generalization of successor representations Dayan (1993) to continuous states, and have a simple linear relationship to successor features Touati & Ollivier (2021). They can therefore be used to both encourage learning a generalizable state representation as well as simultaneously learning the successor features for the BFM. Successor measures are usually learned for an explicitly defined class of policies (Agarwal et al., 2024) or implicitly by first defining a class of reward functions (Touati et al., 2022; Park et al., 2024) and considering optimal policies for those reward functions as the set of policies. The main intuition behind predicting successor measures as a target for state representation learning is that representations sufficient to explain future state-visitation for a wide range of policies capture features that are relevant for sequential decision making under various reward functions.

Unfortunately, learning state representations by estimating successor measures requires iteratively applying Bellman evaluation backups or Bellman optimality backups, both of which are known to result in a variety of learning difficulties. They can suffer from various forms of bias Thrun & Schwartz (2014); Fujimoto et al. (2018); Lu et al. (2018); Fu et al. (2019) and can suffer from feature collapse due to the instability inherent in bootstrapping in the function approximation regime (Kumar et al., 2021). Using Bellman backups to learn a representation requires choosing a class of policies or a class of reward functions a priori. Further, because the state representation is trained from a batch of offline data, unless chosen carefully, the policies may select out-of-distribution actions, leading to incorrect generalization and degenerate representations.

A simple alternative that sidesteps these issues is to use latent dynamics learning: predicting future latent states given the current state and the sequence of actions. Learning the state representation by predicting the latent dynamics has the benefit of being independent of the policy and thus avoids using Bellman backups and these out-of-distribution issues. This work investigates the following question:

*Does latent next-state prediction produce state features that enable performant zero-shot RL?*

Our investigation is inspired by the work of Fujimoto et al. (2025), which showed that using dynamics prediction losses as auxiliary losses boosted performance of a single-task RL agent. Unlike the single task RL setting examined by Fujimoto et al. (2025), we find that in its naive form, this objective leads to a mild form of feature collapse where the representation of different states increase in similarity over training. This collapse results in poor zero-shot RL performance when evaluated on a number of downstream tasks. With a simple orthogonality regularization to prevent collapse, we show that the representations learned are competitive and present a scalable alternative to representations learned via complex successor measure estimation methods for zero-shot RL.

In summary, the contributions of this paper are as follows. 1. We propose regularized latent-dynamics prediction (`RLDP`) as a simple alternative to learn state features for zero-shot RL. We identify as well as mitigate feature collapse plaguing latent dynamics prediction. 2. We show that our method remains competitive through an extensive empirical evaluation of representations for task generalization across a variety of domains, in online and offline RL settings, including in humanoid with a large state-action space. 3. We show that the `RLDP` objective can learn performant policies in low-coverage settings where other methods fail.

## 2 RELATED WORK

**Unsupervised RL** encompasses the class of algorithms that enable learning general-purpose skills and representations without relying on reward signal in the data. Works that have focused on intent or skill discovery have used diversity-driven objectives (Eysenbach et al., 2018; Achiam et al., 2018), maximizing mutual information (Warde-Farley et al. (2018), Eysenbach et al. (2018),

Achiam et al. (2018), Eysenbach et al. (2021)) or minimizing the Wasserstein distance (Park et al. (2023)) between latents and the induced state-visitation distribution. These discovered skills can be used to compose optimal policies for several rewards. Our work, on the other hand, focuses on learning representations capable of producing optimal value functions for any arbitrary function reward specification.

There are also a variety of pre-training approaches for representations that can be fine-tuned for downstream control. Recent pre-training approaches (e.g., Ma et al. (2022); Nair et al. (2022)) borrow self-supervised techniques such as temporal contrastive objectives to extract embeddings from large-scale datasets (Grauman et al. (2021)). HILP (Park et al. (2024)) goes beyond standard masked autoencoding approaches by using Hilbert-space representations to preserve temporal dynamics. Auxiliary objectives involve complementary predictive tasks to get richer semantic or temporal structures (Agarwal et al.; Schwarzer et al., 2020). Although representations from auxiliary objectives can accelerate policy learning, a new policy still needs to be learned from scratch for each new reward function.

**Behavioral Foundation Models** are obtained by training an RL agent in an unsupervised manner using task-agnostic reward-free offline transitions. Forward-Backward representations (Touati & Ollivier (2021)) and PSM (Agarwal et al. (2024)) provide one such framework for training BFMs by learning representations that capture a set of successor measures, on which several successive works are based. Fast Imitation with BFMs (Pirotta et al. (2023)) demonstrates the ability of successor-measure–based BFMs to imitate new behaviors from just a few demonstrations, while Sikchi et al. (2025) builds upon this by fine-tuning the BFM's latent embedding space, yielding 10-40% improvement over their zero-shot performance. Recent progress in imitation learning has led to the development of BFMs tailored for humanoid control tasks (Peng et al. (2022), Won et al. (2022), Luo et al. (2023), Tirinzoni et al. (2025)) which can produce diverse behaviors trained using human demonstration data. Our work differs from these in that it provides a new, simpler state-representation learning objective for training BFMs.

## 3 PRELIMINARIES

We consider a reward-free Markov Decision Process (MDP) (Puterman, 2014) which is defined as a tuple $\mathcal{M} = (\mathcal{S}, \mathcal{A}, P, d_0, \gamma)$, where $\mathcal{S}$ and $\mathcal{A}$ respectively denote the state and action spaces, $P$ denotes the transition dynamics with $P(s'|s,a)$ indicating the probability of transitioning from $s$ to $s'$ by taking action $a$, $d_0$ denotes the initial state distribution and $\gamma \in (0,1)$ specifies the discount factor. A policy $\pi$ is a function $\pi : \mathcal{S} \to \Delta(\mathcal{A})$ mapping a state s to probabilities of action in $\mathcal{A}$. We denote by $\Pr(\cdot \mid s, a, \pi)$ and $\mathbb{E}[\cdot \mid s, a, \pi]$ the probability and expectation operators under state-action sequences $(s_t, a_t)_{t \geq 0}$ starting at $(s, a)$ and following policy $\pi$ with $s_t \sim P(\cdot \mid s_{t-1}, a_{t-1})$ and $a_t \sim \pi(\cdot \mid s_t)$. Given any reward function $r : \mathcal{S} \to \mathbb{R}$, the Q-function of $\pi$ for $r$ is $Q_r^\pi(s,a) := \sum_{t \geq 0} \gamma^t \mathbb{E}[r(s_{t+1}) \mid s, a, \pi]$.

**A Behavioral Foundation Model (BFM) using Successor Features** is a tuple $(\varphi, \psi, \pi_z)$ for state features, $\varphi : \mathcal{S} \to \mathcal{Z}$, successor features defined as $\psi(s, a, \pi) = \mathbb{E}_\pi[\sum_t \gamma^t \varphi(s_t)|s, a]$ and the task-conditioned learned policy $\pi_z$ that inputs any task embedding $z$ that corresponds to a reward function $r_z = \varphi^\top z$. To define this policy $\pi_z$, we use action-values produced by the successor features $\psi$. The action-value function for reward $r_z$ and a fixed policy $\pi$ can be written as

$$Q_z^\pi(s,a) = \mathbb{E}_\pi\Big[\sum_t \gamma^t \varphi(s_t)^\top z|s,a\Big] = \mathbb{E}_\pi\Big[\sum_t \gamma^t \varphi(s_t)^\top|s,a\Big]z = \psi(s,a,\pi)^\top z$$

The policy $\pi_z$ is the optimal policy for reward $r_z$, obtained by iteratively greedifying over $Q_z^\pi(s,a)$. Using the successor feature notation, we can iteratively update $\pi_z$ for all states until it satisfies the following fixed-point equation

$$\pi_z(\cdot|s) = \arg\max_a \psi(s, a, \pi_z)^\top z \quad \text{for all } s. \tag{1}$$

We overload notation and write $\psi(s, a, z)$ to mean $\psi(s, a, \pi_z)$, because later we will directly input $z$ into a network to learn these successor features for a near-optimal policy for $r_z$.

The BFM can be used for any new reward function as long as we can obtain the $z$ corresponding to that reward. This is straightforward to do if we are given a dataset $\rho$, as the corresponding $z$

can be extracted by solving the linear regression problem, $\min_z \mathbb{E}_\rho[(\varphi^\top z - r)^2] = (\varphi^\top \varphi)^{-1} \varphi^\top r$. Naturally, BFMs depend heavily on the choice of the state representation $\varphi$; how to learn an effective $\varphi$ is the subject of this work.

**Learning the successor features** can be done using successor measures, instead of directly estimating the discounted sum of features $\varphi$. The *successor measure* (Dayan, 1993; Blier et al., 2021) of state-action $(s, a)$ under a policy $\pi$ is the (discounted) distribution over future states obtained by taking action $a$ in state $s$ and following policy $\pi$ thereafter:

$$M^\pi(s, a, X) := \sum_{t \geq 0} \gamma^t \mathrm{Pr}(s_{t+1} \in X \mid s, a, \pi) \quad \forall X \subset \mathcal{S}. \tag{2}$$

The action-value can be represented as, $Q^\pi(s, a) = \sum_{s^+} M^\pi(s, a, s^+) r(s^+)$. This simple linear relationship between action-value functions and successor measures is similar to that of successor features and has been exploited by recent works (Touati & Ollivier, 2021; Agarwal et al., 2024; Park et al., 2024) to train BFMs. It has been shown by Touati & Ollivier (2021) that parameterizing the successor measures as $M^{\pi_z}(s, a, s^+) = \psi^\pi(s, a, z)^\top \phi(s^+)$ yields $\psi(s, a, z)$ as successor features for the state feature $\varphi(s) = (\phi\phi^\top)^{-1} \phi(s)$ (Theorem 12 of Touati & Ollivier (2021)). Since, the closed form solution for $z$ for any reward function $r$ was $(\varphi\varphi^\top)^{-1}\varphi r$, using the parameterization of $M^\pi$ implies $z = \phi^\top r$.

**To train the BFM**, we alternate between a successor measure learning phase to get and a policy improvement phase. The successor measure learning phase learns to model densities $M^{\pi_z}(s, a, s^+)$ using the contrastive objective (Blier et al., 2021):

**Successor-measure estimation:** $\quad \mathcal{L}_{SM}(M^{\pi_z}) = -\mathbb{E}_{s,a,s'\sim\rho}[M^{\pi_z}(s, a, s')]$

$$+ \frac{1}{2}\mathbb{E}_{s,a,s'\sim\rho,s^+\sim\rho}[(M^{\pi_z}(s, a, s^+) - \gamma\bar{M}^{\pi_z}(s', \pi_z(s'), s^+))^2]. \tag{3}$$

This objective can be used assuming a fixed state representation $\phi$, only training the successor features $\psi$ (as we will do for our approach), or allows for both $\phi$ and $\psi$ to be jointly optimized, as was done in the Forward-Backward (FB) algorithm (Touati & Ollivier, 2021), PSM (Agarwal et al., 2024) and HILP (Park et al., 2024).

The policy improvement step greedily optimizes the action-value function given by this successor measure

$$\pi_z(s) = \arg\max_a Q^{\pi_z}(s, a) = \arg\max_a \sum_{s^+} M^{\pi_z}(s, a, s^+) \cdot r(s^+)$$

$$= \arg\max_a \sum_{s^+} [\psi(s, a, z)^\top (\phi(s^+) \cdot r(s^+))]$$

$$= \arg\max_a \psi(s, a, z)^\top \sum_{s^+} \phi(s^+) \cdot r(s^+) = \arg\max_a \psi(s, a, z)^\top z \tag{4}$$

In practice, we cannot directly set the policy to this argmax. Instead, we optimize the following loss.

**Policy Improvement:** $\qquad \mathcal{L}_P(\pi_z) = -\mathbb{E}_{a\sim\pi_z(s)}[\psi(s, a, z)^\top z] \tag{5}$

Appendix A.2 provides a further overview of approaches to train BFMs. In this work, we leverage this machinery for BFMs and focus on a new approach to estimate the state representations $\phi$.

## 4 METHOD

This method can be broadly divided into two parts: representation learning and zero-shot RL using successor features. The state representation encoder is trained using latent dynamics prediction with diversity regularization. We will show that these representations lead to a reduction in the prediction error for successor measures for any policy. Leveraging these robust state embeddings, we then pretrain a Behavioral Foundation Model (BFM) to predict successor measures, enabling zero-shot inference of near-optimal policies for unseen reward functions. We refer to this method as RLDP (**R**egularized **L**atent **D**ynamics **P**rediction based Behavioral Foundation Policies).

## 4.1 LEARNING REPRESENTATIONS WITH REGULARIZED LATENT DYNAMICS PREDICTION

Zero-shot RL based on successor features relies on learning a state representation denoted by $\phi(s)$. This state representation will define the span of reward functions that the zero-shot RL method is guaranteed to output optimal policies for.

The primary representation learning objective is unrolled latent dynamics prediction. We learn a state representation encoder $\phi : \mathcal{S} \to \mathbb{R}^d$, $(\mathcal{Z} = \mathbb{R}^d)$ and a latent state-action representation encoder $g : \mathbb{R}^d \times \mathcal{A} \to \mathbb{R}^d$ such that latent dynamics can be expressed as $\phi(s') = g(\phi(s), a)^\top w$ with some learned weights $w$, informing our loss function for representation learning. A sub-sequence of horizon $H$ is sampled from the offline interaction dataset $\rho$ given by $\tau^i = \{s_0^i, a_0^i, s_1^i, a_1^i, ..., s_{H-1}^i, a_{H-1}^i, s_H^i\}$. A sequence of future latent states $h_{1:H}^i$ are obtained by encoding the initial state $h_0^i = \phi(s_0^i)$ and unrolling using the defined dynamics model $h_{t+1}^i = g(h_t^i, a_t)^\top w$. Then the objective is to predict the encoded future latent states:

$$\mathcal{L}_d(\phi, g, w) = \mathbb{E}_{\tau^i \sim d^O}\left[\left\|\sum_{t=1}^{H} h_t^i - \bar{\phi}(s_t^i)\right\|^2\right], \tag{6}$$

where $\tau^i = \{s_0^i, a_0^i, s_1^i, a_1^i, ..., s_{H-1}^i, a_{H-1}^i, s_H^i\}$, $h_0^i = \phi(s_0^i)$, and $\bar{\phi}$ is the slowly moving encoder target, which is periodically set to $\phi$.

Latent dynamics models have been shown to significantly improve sample efficiency for single task RL when models are used for planning (Hansen et al., 2022), learning (Hafner et al., 2019), or as representations (Fujimoto et al., 2025) for model-free RL, but their suitability as general-purpose representations for multi-task and zero-shot RL remains understudied. Most successful methods (Touati & Ollivier, 2021; Agarwal et al., 2024) for zero-shot RL train representations to predict successor measures. However, directly estimating successor measures requires learning future state-occupancies under a predefined set of policies. This poses a problem in the low-coverage setting as Bellman backups with policies that choose out of distribution action

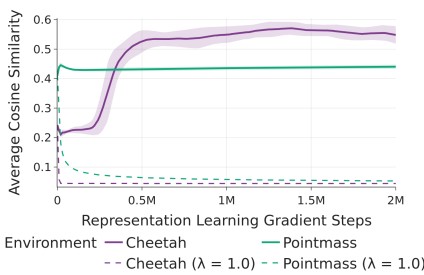

**Figure 1:** Average Cosine similarity between state-representations sampled uniformly from the training dataset. Feature similarity increases over the course of training; once adding our orthogonality regularizer (with $\lambda = 1$), we obtain more diverse representations. Shaded region shows standard deviation over 4 seeds.

will result in incorrect predictions and negatively affect representation learning. In contrast, latent dynamics prediction is a policy-independent representation learning objective.

However, solely learning with the latent dynamics objective can lead to convergence to a collapsed solution. This is unsurprising as trivial solutions of predicting a constant zero vector achieves a perfect loss in equation 6. To combat this, prior works (Grill et al., 2020) have proposed the use of a semi-gradient update where a stop-gradient is used for target $h_{t+1}$ in equation 6 along with a slowly updating target. However, we find these techniques insufficient to maintain representation diversity. We investigate this by computing the cosine similarity of state representations as a function of gradient steps trained via minimizing equation 6 on an offline dataset collected by an exploration algorithm RND (Burda et al., 2018). Figure 1 shows that while the solutions do not collapse, there is an increase in feature similarity over the course of learning, which we refer to as a *mild* form of collapse. As the space of reward functions is spanned by state features, such an increase in feature similarity directly reduce the class of reward functions for which we can learn optimal policies and negatively impact task generalization.

**Mitigating collapse in latent dynamics prediction:** In order to prevent the mild form of feature collapse discussed earlier, we propose to add an auxiliary regularization objective that encourages diversity. Orthogonal regularization has been also studied in self-supervised learning (He et al., 2024; Bansal et al., 2018) as a way to mitigate collapse. We project all state representations $\phi$ as well as predicted latent next-state $g(\phi(s), a)^\top w$ in a hypersphere: $\mathbb{S}^{d-1} = \{x \in \mathbb{R}^d : \|x\|_2 = \sqrt{d}\}$ and regularize by minimizing cosine similarity between any two states. We

ablate the choice of hyperspherical normalization on $g$ in Appendix A.4.2 and observe it to give consistent improvements. We note that a similar regularization was applied to state features in the implementation for Forward-Backward representations (Touati et al., 2022) to encourage solution identifiability and uniqueness. In the case of latent dynamics prediction this step becomes crucial to mitigate the increase in representation similarity.

The orthogonal regularization loss takes the following form:

$$\mathcal{L}_r(\phi) = \mathbb{E}_{s,s'\sim\rho}[\phi(s)^\top \phi(s')] \qquad (7)$$

where $\phi \in \mathbb{S}^{d-1}$. Our final loss is a weighted combination of dynamics prediction combined with orthogonal diversity regularization

$$\mathcal{L}_{RLDP}(\phi, g, w) = \mathcal{L}_d(\phi, g, w) + \lambda \mathcal{L}_r(\phi) \quad (8)$$

where $\lambda$ controls the regularization strength. We visualize this loss in figure 2, where the encoder is given by $\phi$, the dynamics by $g$ and $w$ and the diversity across encoded states is encouraged with $\mathcal{L}_r$. We find that adding this regularization prevents collapse, shown in figure 1, even with a relatively small regularization coefficient $\lambda = 0.01$. We evaluate the impact of orthogonality regularization further with other coefficients $\lambda$ in Appendix section A.4.2.

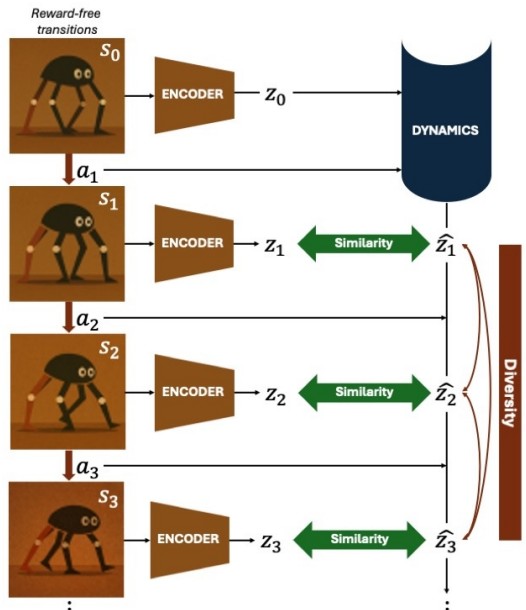

**Figure 2:** `RLDP` combines latent next state prediction + regularization for diversity (an orthogonality regularizer) to learn representations for BFMs.

**RLDP leads to representations capable of predicting successor measures:** The representation learning objective is simply latent dynamics prediction with an orthogonal regularization. Through this objective, we are enforcing the representations to be good for predicting successor measures, which forms the basis of the BFMs that will be constructed using these. While prior work (Agarwal et al., 2025) already indicate that these representations are suitable for predicting successor measures, we further formalize this intuition. Let's begin by looking at the latent space abstract MDP $\bar{\mathcal{M}}$ defined using the state representation $\phi$.

**Definition 4.1.** *Let MDP $\bar{\mathcal{M}}$ corresponding to the state abstraction $\phi : \mathcal{S} \to \mathcal{Z}$ be defined as $< \phi(s), \mathcal{A}, P(\cdot|\phi(s), a), \gamma, r >$.*

Apart from facilitating the construction of BFMs and zero-shot RL, one of the utilities of the state-representations is to compress the state space to a smaller space. MDP $\bar{\mathcal{M}}$ represents this compression. We will assume that $\bar{\mathcal{M}}$ is Lipschitz. Formally,

**Assumption 4.2.** *$\bar{\mathcal{M}}$ is $(\mathcal{K}_R, \mathcal{K}_P) - Lipschitz.$ (Gelada et al., 2019)*

We now have all the components to show in the Lemma 4.3 that minimizing $\mathcal{L}_{RLDP}$ will lead to a reduction in the prediction error of successor measure for any $\mathcal{K}_V$-Lipschitz valued policy.

**Lemma 4.3.** *Given MDP $\bar{\mathcal{M}}$, let $\pi$ be any $\mathcal{K}_V$-Lipschitz valued policy, $M^\pi$ be the successor measure for $\pi$ and $\bar{M}^\pi$ be the corresponding successor measure on $\bar{\mathcal{M}}$, $\mathcal{L}_{RLDP}(\phi, g, w)$ upper bounds the prediction error in successor measure,*

$$\mathbb{E}_{s,a\sim d^\pi, s^+\sim\rho}[|M^\pi(s, a, s^+) - \bar{M}^\pi(\phi(s), a, \phi(s^+))|] \le \frac{\mathcal{L}_{RLDP}(\phi, g, w)}{1 - \gamma} \qquad (9)$$

## 4.2 Zero-shot RL with RLDP Representations

The first step to use `RLDP` is to train the representations from a batch of reward-free offline environment transitions. The `RLDP` representation loss given in equation 8 does not rely on reward, because it only uses the latent dynamics prediction loss and the orthogonality regularizer.

It is straightforward to optimize this loss on the batch of offline data, to obtain a learned state representation $\phi$. Key choices to be made include the regularization coefficient $\lambda$, the length of the rollout $H$ for the latent dynamics prediction loss and the update frequency for encoder target $\bar{\phi}$.

The next step is to train the BFM, by alternating successor measure estimation and policy improvement. The `RLDP` representations are kept frozen in the successor measure parameterization $M^{\pi_z}(s, a, s^+) = \psi^\pi(s, a, z)^\top \phi(s^+)$ and $\psi(s, a, z)$ and $\pi_z$ are trained using Eq. 10 and Eq. 5 respectively.

$$\mathcal{L}_{zsrl}(\psi) = -\mathbb{E}_{s,a,s'\sim\rho}[\psi(s, a, z)\phi(s')]$$
$$+ \frac{1}{2}\mathbb{E}_{s,a,s'\sim\rho,s^+\sim\rho}[(\psi(s, a, z)\phi(s^+) - \gamma\bar{\psi}(s', \pi_z(s'), z)\phi(s^+))^2] \quad (10)$$

Following prior work, in our experiments we consider variations of the policy improvement step (Eq 5) where we use an expert regularization in the policy update (Tirinzoni et al., 2025) to guide exploration during online RL for high-dimensional state-action spaces or use a behavior cloning regularization (Fujimoto & Gu, 2021) when learning offline for low-coverage datasets. These modifications are discussed in detail in the next section. We provide the full representation and policy learning pipeline for `RLDP` in Appendix section A.6.

## 5 EXPERIMENTS

The goal of our experiments is to perform an extensive empirical study of the suitability of state representations learned by a regularized latent next-state prediction objective when compared to other methods that employ more complex strategies. In particular, we aim to answer the following questions: (a) Keeping all other learning factors similar, how does our method compare to baselines in enabling generalization to unseen reward functions? We compare the representations learned by training multi-task policies with zero-shot RL both in the offline setting and the online setting. (b) By avoiding querying actions out of distribution does `RLDP` provide a robust choice for learning representations in low coverage datasets? (c) What design decisions are crucial to the success of our method? We perform extensive ablation studies to understand our design choices.

For all datasets, we pretrain a BFM using the successor feature approach outlined in our method section 4. Each algorithm is given the same budget of gradient steps during pretraining, controlling the state representation dimension, and the final performance is obtained by taking the pre-trained model at the end and querying it for different task-rewards for 50 episodes.

### 5.1 BENCHMARKING ZERO-SHOT RL FOR CONTINUOUS CONTROL

**Baselines:** We broadly compare `RLDP` against commonly used state-of-the-art baselines for zero-shot RL such as: FB, PSM, and HILP. These baselines represent a set of diverse and strong approaches in the area of zero-shot RL.

#### 5.1.1 OFFLINE ZERO-SHOT RL

**Setup:** We consider continuous control tasks from DeepMind control suite (Tassa et al., 2018) – Pointmass, Cheetah, Walker, Quadruped under a similar setup considered

| | Task | Random Features | FB | PSM | RLDP |
|---|---|---|---|---|---|
| **Walker** | Stand | 392.40±58.03 | **918.29±28.83** | **899.54±30.73** | **890.40±27.33** |
| | Run | 75.39±20.97 | 381.31±17.32 | **450.57±28.95** | 334.26±49.69 |
| | Walk | 193.84±112.98 | 779.29±63.60 | **875.61±33.44** | **779.77±137.16** |
| | Flip | 132.02±67.85 | **977.08±2.76** | 621.36±75.62 | 492.94±22.79 |
| **Cheetah** | Run | 31.82±36.88 | 129.39±37.63 | **181.85±54.17** | 157.12±29.92 |
| | Run Backward | 60.08±12.82 | 142.41±36.77 | **158.64±18.56** | **170.52±15.30** |
| | Walk | 147.52±155.66 | **604.54±80.51** | 576.98±209.45 | 592.92±104.66 |
| | Walk Backward | 272.77±42.40 | 630.40±144.23 | **817.92±98.86** | **821.51±50.62** |
| **Quadruped** | Stand | 240.01±66.06 | 732.59±101.33 | 708.03±34.99 | **794.94±43.25** |
| | Run | 114.19±30.22 | 425.15±52.02 | 404.32±23.26 | **457.41±74.70** |
| | Walk | 137.65±47.57 | 492.91±17.55 | **523.94±52.13** | 465.40±185.29 |
| | Jump | 190.62±46.63 | 567.27±48.90 | 549.57±15.86 | **733.32±55.30** |
| **Pointmass** | Top Left | 258.59±183.56 | **943.85±17.31** | 924.20±10.64 | 890.41±60.79 |
| | Top Right | 216.30±189.05 | **550.84±282.41** | 666.00±133.15 | 795.47±21.10 |
| | Bottom Left | 193.32±90.37 | 672.28±153.06 | **800.93±15.62** | 805.17±20.44 |
| | Bottom Right | 64.08±72.21 | **272.97±274.99** | 123.44±138.82 | 193.38±167.63 |

**Table 1:** Comparison (over 4 seeds) of zero-shot RL performance between using an untrained initialized encoder, FB, PSM, and `RLDP` with representation size $d = 512$. Bold indicates the best mean and any method whose mean plus one standard deviation overlaps with the best mean.

by prior works in zero-shot RL. We use datasets from the ExoRL suite (Yarats et al. (2022)) that are obtained by an exploratory algorithm RND (Burda et al. (2018)). Random features use representations from a randomly initialized NN encoder.

**Evaluation:** To evaluate the different zero-shot RL methods we take the pretrained policies and query them on a variety of tasks. For each environment, we consider 4 tasks similar to prior works (Touati et al., 2022; Park et al., 2024; Agarwal et al., 2024). We conduct our experiments across two axes: a) Table 1 pretrains all the BFMs on same number of representation dimensions (512) and gradient steps. For RLDP, we use an encoding horizon of 5. We train representations for 2 million steps and train policy for additional 3 million steps. b) Table 6 in the Appendix compares against representation dimension for $\phi$ found to be best for prior methods and RLDP with the same number of gradient updates for pretraining each BFM.

**Results:** Overall, using learned representations (FB, PSM, RLDP) outperforms random features, confirming that representation learning is crucial for zero-shot RL. Among learned methods, PSM and RLDP generally achieve the strongest performance. Furthermore, training FB and PSM baselines is sensitive to hyperparameters and we rely on author's implementation to tune hyperparameters.

### 5.1.2 ONLINE ZERO-SHOT RL

Previous section validated that RLDP representations lead to competitive zero-shot RL when the learning policies use offline interaction data. We explore if the learned representation enable competitive multi-task learning when agent is allowed interaction with the environment.

**Setup:** We consider the SMPL (Loper et al. (2015)) Humanoid environment that aims to mimic real human embodiment and provides a complex learning challenge with a 358 dimensional observation space and a 69 dimensional action space. Due to the exploratory challenge of the environment, Tirinzoni et al. (2025) presented a new approach, Conditional Policy Regularization (CPR), to guide RL learning regularized with expert real-human trajectories. CPR trains successor measures in a similar way as equation 3 but adds a regularization objective to policy encouraging it to jointly maximize $Q$-function while staying close to expert. This allows for

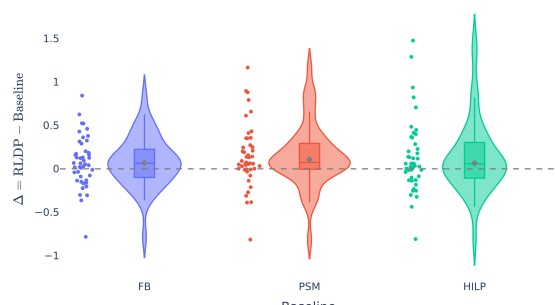

**Figure 3:** Pair-wise comparison of RLDP against prior offline representation learning methods using per-task oracle normalized performance differences ($\Delta$ = RLDP – Baseline) in SMPL Humanoid environment. The gray diamond represents the IQM (Interquartile Mean).

better exploration and more realistic motions. Further implementation details can be found in Appendix sections A.3.2, A.5.2.

**Evaluation:** Our representation learning phase is offline and we use the metamotivo 5M transition dataset[1] collected from replay buffer of an online RL agent to learn state-representations and then use the CPR approach to train zero-shot policies. We train representations for 2 million gradient steps and policy for 20 million environment steps. The offline phase of representation learning helps us remove the exploration confounder and test the quality of representations obtained by different approaches. The evaluation is performed on the full suite of 45 tasks provided by Tirinzoni et al. (2025). For each task, we present the normalized scores with respect to fully-online trained representations and policy in table 8 and we present the aggregates results across tasks in figure 3 over 4 seeds.

**Results:** Figure 3 summarizes the results of RLDP representations with respect to baseline methods across all 45 tasks. Positive values indicate tasks where RLDP achieves higher normalized returns. These results suggest that overall RLDP fares competitively to the baselines. Complete results for this evaluation are provided in table 8. Further analysis shows that the performance is task dependent - on some tasks (such as raisearms and lieonground), RLDP outperforms the baselines, even beating

---

[1]https://huggingface.co/facebook/metamotivo-M-1

the oracle performance for some tasks (shown in table 8). In others (like crawl or rotate tasks), all methods perform subpar to oracle.

## 5.2 LEARNING REPRESENTATIONS WITH LOW COVERAGE DATASETS

`RLDP` learns a policy-independent representation through latent dynamics prediction. Prior approaches assume a class of policies to learn representations predictive of successor measures, and this strategy can lead to poor out-of-distribution generalization when actions proposed by the policy are not covered by the dataset.

**Setup.** To evaluate this hypothesis concretely, we consider the D4RL benchmark of OpenAI Gym MuJoCo tasks (Fu et al. (2020), Todorov et al. (2012), Brockman et al. (2016)). This dataset has been widely used to examine the effects of value estimation error from out-of-distribution actions due to low coverage, which many offline RL algorithms struggle with (Kostrikov et al. (2021); Fujimoto & Gu (2021); Kumar et al. (2020); Wu et al. (2019); Sikchi et al. (2023)). We consider halfcheetah, hopper, and walker2d domains, and medium and medium-expert datasets.

**Evaluation:** To evaluate the different zero-shot RL methods, we first pretrain the representation learning methods on these datasets for 1 million gradient steps. We use a modified zero-shot policy learning approach that alternates between equation 3 and equation 5 that is additionally augmented policy improvement loss with a behavioral regularization inspired by Fujimoto & Gu (2021). This regularization allow the RL approach to learn without overestimation bias and enabling us to establish a fair comparison among representations learned by different approaches. We use the corresponding reward function provided by each dataset to do reward inference and evaluate the zero-shot policy. Further details are provided in Appendix sections A.3.3 and A.5.3.

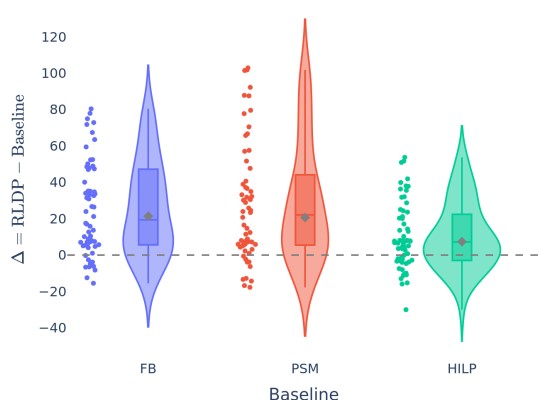

**Figure 4:** Pair-wise comparison of `RLDP` against baseline representation learning methods in low-coverage D4RL dataset. Each point represents $\Delta = R_{\text{RLDP}} - R_{\text{baseline}}$ for a single $\{\text{task}, \text{seed}\}$ pair. The gray diamond represents the IQM (Interquartile Mean).

**Results:** Figure 4 shows one-to-one comparison of normalized returns of `RLDP` against baseline methods using paired per-seed performance differences across 6 low-coverage D4RL tasks over 10 seeds. `RLDP` outperforms all baseline methods in 5 out of 6 tasks. The overall mass of the violin lies above zero and IQM is positive, indicating that `RLDP` achieves higher normalized returns compared to the baseline. Overall, the results suggest that `RLDP` is a reliable choice for feature learning in low coverage datasets while providing a simpler alternative to otherwise complex representation learning approaches. Per task normalized scores and statistical significance testing is reported in the Appendix section A.9 and table 9. We further bisect the individual and combined impact of using Bellman backups (similar to FB which may query out-of-distribution actions), latent next-state prediction, and orthogonality regularization for representation learning in Appendix section A.11 and find explicit Bellman backups to hurt performance of learned policy.

## 5.3 WHAT MATTERS FOR SUPERVISING REPRESENTATIONS SUITABLE FOR CONTROL?

In section 4, we introduced `RLDP` method of representation learning with the loss used (equation 8) and the encoder training process. In this section, we aim to ablate components of this loss and the architecture of the encoder.

**Orthogonality regularization:** Keeping the encoding horizon constant ($H = 5$), we change the orthogonality regularization coefficient. The results, presented in figure 5, show that for zero regularization ($\lambda = 0$), the average return decreases compared to $\lambda > 0$. This shows that

diversity regularization is critical to the representation loss. For fixed encoding horizon, we see that orthogonality regularizer $\lambda = 1$ performs best. To further understand the role of the orthogonality regularizer in representation learning and how it helps prevent feature collapse, we refer to section 4 and Appendix section A.4.2, where we show that the learned representations increase in cosine similarity without regularization.

**Encoder architecture:** In section 4.1, we introduce encoder training, where we project the latent next state representation $g(\phi(s), a).w$ to a hypersphere. Here, we ablate the importance of this projection. The results are presented in table 2. We observe that RLDP consistently outperforms its variant without spherical normalization on most tasks. The standard deviation is also higher for most results on the variant without hypersphere projection. This indicates that spherical normalization is an important design choice for stabilization and improving performance.

Results for all environments are reported in table 5. We provide

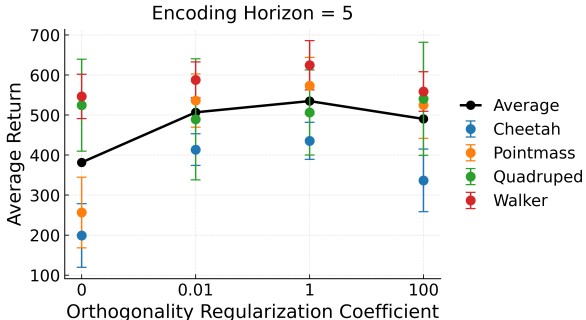

**Figure 5:** Evaluating the impact of Orthogonality Regularization: We ran one-sided Mann–Whitney U tests on the per-seed returns over 4 seeds to compare different values of the orthogonality regularization, and we observe that adding small orthogonality regularization coefficient $\lambda = 0.01$ gives a statistically significant improvement over coefficient $\lambda = 0.0$.

additional discussion of the complete encoder architecture in Section A.4, where we further ablate the encoder architecture.

## 6 CONCLUSION

This paper introduces RLDP, a representation learning objective for effective task generalization enabling performant behavioral foundation models. Our objective takes the simple form of regularized latent-dynamics prediction, an objective that does not require any reconstruction, making it able to handle high-dimensional observation space and does not require explicit Bellman backups, making it more amenable to optimization. We identify that simply using latent-dynamics prediction leads to a mild form of feature collapse where the state-representation similarity increases over

| Task | | RLDP | RLDP w/o SN |
|---|---|---|---|
| **Quadruped** | Stand | 794.94±43.25 | 661.73±95.75 |
| | Run | 457.41±74.70 | 378.97±148.47 |
| | Walk | 465.40±185.29 | 519.39±251.11 |
| | Jump | 733.32±55.30 | 495.98±133.81 |
| | **Average(*)** | 612.77±83.17 | 514.02±86.94 |
| **Pointmass** | Top Left | 890.41±60.79 | 892.13±41.74 |
| | Top Right | 795.47±21.10 | 728.72±122.99 |
| | Bottom Left | 805.17±20.44 | 683.12±76.22 |
| | Bottom Right | 193.38±167.63 | 22.54±39.04 |
| | **Average(*)** | 671.11±292.58 | 581.63±341.00 |

**Table 2:** Study of encoder architecture (subset). Table shows mean ± std; RLDP significantly outperforms RLDP w/o SN on Pointmass, Quadruped, and pooled. SN: Spherical Normalization on $g$.

time. To combat this issue, we propose using orthogonal regularization as a way to maintain feature diversity and prevent collapse. Using our method enables learning generalizable, stable, and robust representations that can achieve competitive performance compared to prior zero-shot RL techniques without relying on reinforcement-driven signals. Importantly, we show that prior approaches struggle in low coverage setting and RLDP works robustly across different dataset types, making it a practical unsupervised learning approach. This work, thus, paves the way for simpler yet effective approaches to learn zero-shot policies in behavioral foundation models.

## 7 ACKNOWLEDGEMENTS

We thank Siddarth Chandrasekar, Dikshant Shehmar, and Diego Gomez for enlightening discussions on unsupervised RL. This work has been conducted at the Reinforcement Learning and Artificial Intelligence (RLAI) lab at the University of Alberta, the Safe, Correct, and Aligned Learning and Robotics Lab (SCALAR) at the University of Massachusetts Amherst, and Machine Intelligence through Decision-making and Interaction (MIDI) Lab at The University of Texas at Austin. Support for this work was provided by the Canada CIFAR AI Chair Program, the Alberta Machine Intelligence Institute, and the Natural Sciences and Engineering Research Council of Canada (NSERC). HS, SA, and AZ are supported by NSF 2340651, NSF 2402650, DARPA HR00112490431, and ARO W911NF-24-1-0193. We are also grateful for the computational resources provided by the Digital Research Alliance of Canada.

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

# A   APPENDIX

## A.1   PROOF OF LEMMA 4.3

**Lemma A.1.** *Given MDP $\bar{\mathcal{M}}$, let $\pi$ be any $\mathcal{K}_V$-Lipschitz valued policy, $M^\pi$ be the successor measure for $\pi$ and $\bar{M}^\pi$ be the corresponding successor measure on $\bar{\mathcal{M}}$, $\mathcal{L}_{RLDP}(\phi, g, w)$ upper bounds the prediction error in successor measure,*

$$\mathbb{E}_{s,a\sim d^\pi,s^+\sim\rho}[|M^\pi(s,a,s^+) - \bar{M}^\pi(\phi(s),a,\phi(s^+))|] \leq \frac{\mathcal{L}_{RLDP}(\phi,g,w)}{1-\gamma} \quad (9)$$

*Proof.* Let's begin with $\mathbb{E}_{s,a\sim d^\pi}[|M^\pi(s,a,s^+) - \bar{M}^\pi(s,a,s^+)|]$. For a fixed $s^+$

$$\mathbb{E}_{s,a\sim d^\pi}[|M^\pi(s,a,s^+) - \bar{M}^\pi(s,a,s^+)] \leq \mathbb{E}_{s,a\sim d^\pi}|p(s'=s^+) - p(\phi(s')=\phi(s^+))|+$$
$$\gamma\mathbb{E}_{s,a\sim d^\pi}|\mathbb{E}_{s'\sim P(\cdot|s,a)}V^\pi(s') - \mathbb{E}_{\phi(s')\sim P(\cdot|\phi(s),a)}V^\pi(\phi(s')|$$
$$\leq \mathcal{L}_R + \gamma\mathbb{E}_{s,a\sim d^\pi}|\mathbb{E}_{s'\sim P(\cdot|s,a)}[V^\pi(s') - V^\pi(\phi(s'))]|+$$
$$\gamma\mathbb{E}_{s,a\sim d^\pi}|\mathbb{E}_{s'\sim P(\cdot|s,a),\ \phi(s')\sim P(\cdot|\phi(s),a)}[V^\pi(s') - V^\pi(\phi(s'))]||$$
$$\leq \mathcal{L}_R + \gamma\mathbb{E}_{s,a\sim d^\pi}|\mathbb{E}_{s'\sim P(\cdot|s,a)}[V^\pi(s') - V^\pi(\phi(s'))]|+$$
$$\gamma\mathcal{K}_V\mathbb{E}_{s,a\sim d^\pi}D(\phi P(\cdot|s,a), P(\cdot|\phi(s),a))$$
$$(D \text{ is the distance metric used by } \mathcal{K}_V\text{-Lipschitz policy})$$
$$= \mathcal{L}_R + \gamma\mathbb{E}_{s,a\sim d^\pi}|\mathbb{E}_{s'\sim P(\cdot|s,a)}[V^\pi(s') - V^\pi(\phi(s'))]|+$$
$$\gamma\mathcal{K}_V\mathbb{E}_{s,a\sim d^\pi}\mathcal{L}_d$$
$$\leq \mathcal{L}_R + \gamma\mathbb{E}_{s,a\sim d^\pi}\mathbb{E}_{s'\sim P(\cdot|s,a)}|V^\pi(s') - V^\pi(\phi(s')|+$$
$$\gamma\mathcal{K}_V\mathbb{E}_{s,a\sim d^\pi}\mathcal{L}_d$$
$$\leq \mathcal{L}_R + \gamma\mathbb{E}_{s,a\sim d^\pi}|V^\pi(s) - V^\pi(\phi(s))|+$$
$$\gamma\mathcal{K}_V\mathcal{L}_d$$
$$= \mathcal{L}_R + \gamma\mathbb{E}_{s,a\sim d^\pi}|M^\pi(s,a,s^+) - M^\pi(\phi(s),a,\phi(s^+))|+$$
$$\gamma\mathcal{K}_V\mathcal{L}_d$$

This implies, $(1-\gamma)\mathbb{E}_{s,a\sim d^\pi}[|M^\pi(s,a,s^+) - \bar{M}^\pi(s,a,s^+)] \leq \mathcal{L}_R + \gamma\mathcal{K}_V\mathcal{L}_d$ where $\mathcal{L}_R = \mathbb{E}_{s,a\sim d^\pi}|p(s'=s^+) - p(\phi(s')=\phi(s^+))|$.

Taking expectation under $s^+ \sim \rho(s^+)$,

$$(1-\gamma)\mathbb{E}_{s,a\sim d^\pi,s^+\sim\rho}[|M^\pi(s,a,s^+) - \bar{M}^\pi(s,a,s^+)] \leq \mathbb{E}_{s,a\sim d^\pi,s^+\sim\rho}|p(s'=s^+) - p(\phi(s')=\phi(s^+))| + \gamma\mathcal{K}_V\mathcal{L}_d$$
$$= \mathcal{L}_R + \gamma\mathcal{K}_V\mathcal{L}_d$$

This implies, $\mathbb{E}_{s,a\sim d^\pi,s^+\sim\rho}[|M^\pi(s,a,s^+) - \bar{M}^\pi(s,a,s^+)] \leq \frac{\mathcal{L}_r + \gamma\mathcal{K}_V\mathcal{L}_d}{1-\gamma} = \frac{\mathcal{L}_{RLDP}}{1-\gamma}$ $\qquad\square$

## A.2   PRIOR APPROACHES FOR REPRESENTATION LEARNING IN BFM'S

Prior work has often relied on complex objectives to enable learning of $\phi$ and $\psi$ for BFMs. Forward-Backward (FB) (Touati et al., 2022) combine learning the state representation, $\phi$ with successor features, $\psi$ and the policy. $\phi$ and $\psi$ are jointly learned to represent successor measures for a class of reward-optimal policies. FB alternates minimizing the successor measure loss below jointly for $\psi, \phi$ alongside policy improvement by optimizing Eq 5. FB uses the following loss minimizing Bellman residuals to learn representations:

$$\mathcal{L}(\phi,\psi) = -\mathbb{E}_{s,a,s'\sim\rho}[\psi(s,a,z)^T\phi(s')]$$
$$+ \frac{1}{2}\mathbb{E}_{s,a,s'\sim\rho,s^+\sim\rho}[(\psi(s,a,z)^T\phi(s^+) - \gamma\bar{\psi}(s',\pi_z(s'),z)^T\bar{\phi}(s^+))^2] \quad (11)$$

HILP (Park et al., 2024) learns state representation $\phi$ that are suitable to predict value function for goal-reaching which is subsequently used for zero-shot RL in the same way as RLDP. HILP parameterizes the value function to be $V(s, g) = \|\bar{\phi}(s) - \bar{\phi}(g)\|$ and then minimizes:

$$\mathcal{L}(\phi) = \mathbb{E}_{s,s',g\sim\rho}[\ell_\tau^2(-\mathbb{1}(s \neq g) - \gamma V(s', g) + V(s, g)] \tag{12}$$

where $\ell_\tau^2$ is an expectile loss (Kostrikov et al. (2021)).

PSM (Agarwal et al., 2024) learns state representation to represent the successor measures for a class of policies defined with a discrete codebook. The loss used is as follows -

$$\mathcal{L}(\phi, \psi, w) = -\mathbb{E}_{s,a,s'\sim\rho}[\psi(s,a)w(c)\phi(s')]$$
$$+ \frac{1}{2}\mathbb{E}_{s,a,s'\sim\rho,s^+\sim\rho}[(\psi(s,a)w(c)\phi(s^+) - \gamma\bar{\psi}(s', \pi_c(s'))\bar{w}(c)\bar{\phi}(s^+))^2] \tag{13}$$

where $c$ is a discrete code defining a policy and $w$ maps the discrete code to a continuous space. The above loss is minimized averaged over a pre-determined distribution of discrete codes.

The Laplacian approach (Wu et al., 2018) is action-independent and learns state representation using eigenvectors of graph-Laplacian induced by a random-walk operator. The representation objective for Laplacian approach takes the following form:

$$\mathcal{L}(\phi) = \frac{1}{2}\mathbb{E}_{s\sim\rho,\ s'\sim P_\pi(\cdot|s)}\left[\|\phi(s) - \phi(s')\|_2^2\right] + \beta\mathbb{E}_{s,s'\in\rho}[\phi(s)^T\phi(s')] \tag{14}$$

### A.3 EXPERIMENTAL DETAILS

#### A.3.1 EXORL

ExoRL (Exploratory Offline Reinforcement Learning) is a benchmark suite that provides large, diverse offline datasets generated by exploratory policies across multiple domains (e.g., locomotion, manipulation, navigation). We consider three locomotion and one goal-based navigation environments – Walker, Quadruped, Cheetah, Pointmass – from the Deepmind Control Suite (Tassa et al. (2018)). For offline training, we use data provided from the EXORL benchmark trained using RND agent. These domains are explained further in table 3. All DM control tasks have an episode length of 1000.

| Domain | Description | Type | Observation/Action Dimension | Tasks | Reward |
|--------|-------------|------|------------------------------|-------|--------|
| Walker | two-legged robot | Locomotion | 24/6 | stand
walk
run
flip | Dense |
| Quadruped | four-legged robot | Locomotion | 78/12 | jump
walk
run
stand | Dense |
| Cheetah | planar, 2D robot | Locomotion | 17/6 | walk
run
walk backward
run backward | Sparse |
| Pointmass | navigation in 2D plane | Goal-reaching | 4/2 | reach top left
reach top right
reach bottom right
reach bottom left | Sparse |

Table 3: **ExoRL dataset summary**. *Domain* is the environment name in the ExoRL benchmark. *Description* is a natural language description of the agent embodiment and environment. *Type* refers to the broad task category. *Observation/Action Dimension* refers to the size of observation and action vectors from the environment. *Tasks* refers to the suite of evaluation tasks provided in the ExoRL benchmark. *Reward* refers to the density of non-zero reward signals from the environment.

### A.3.2   SMPL 3D HUMANOID

SMPL (Skinned Multi-Person Linear Model) is a 3D parametric model of the human body that is widely used for character animation. It has a 358 dimensional proprioceptive observation space that includes body pose, rotation, and velocities. The action space is 69 dimensional where each action dimension lies in [-1,1]. All episodes are of length 300.

### A.3.3   D4RL

D4RL (Datasets for Deep Data-Driven Reinforcement Learning) (Fu et al. (2020)) is an offline RL benchmark suite built on the v2 Open AI Gym (Brockman et al. (2016)) that provides standardized datasets and evaluation protocols across simulated and real-world tasks. We consider three simulated locomotion tasks – Hopper, HalfCheetah,

| Domain | Task Name | # Samples |
|--------|-----------|-----------|
| Gym-MuJoCo | hopper-medium | $10^6$ |
| | hopper-medium-expert | $2 \times 10^6$ |
| | halfcheetah-medium | $10^6$ |
| | halfcheetah-medium-expert | $2 \times 10^6$ |
| | walker2d-medium | $10^6$ |
| | walker2d-medium-expert | $2 \times 10^6$ |

**Table 4:** Gym-MuJoCo tasks from D4RL.

Walker2D – and two datasets – medium and medium-expert. As described in Fu et al. (2020), the medium dataset is generated by online training a Soft-Actor Critic (Haarnoja et al. (2018)) agent, early-stopping the training, and collecting 1 million samples from this partially-trained policy. The "medium-expert" dataset is generated by mixing equal amounts of expert demonstrations and suboptimal data, generated via a partially trained policy or by unrolling a uniform-at-random policy. Further details about these tasks have been provided in table 4. Episodes have inconsistent length depending on termination/truncation with a maximum of 1000.

### A.4   REGULARIZED LATENT DYNAMICS PREDICTION

RLDP aims to learn a state representation encoder $\phi$ such that latent state dynamics can be expressed as $\phi(s') = g(\phi(s), a)^\top \mathbf{w}$ where $g$ is a latent-state action encoder and w are some constant weights.

### A.4.1   ARCHITECTURE

The architecture of the RLDP latent next-state prediction network is as pictured in figure 6

The state representation network $\phi$ is a feedforward MLP with two hidden layers of 256 units that maps a state $s$ to a $d$-dimensional embedding. In our default RLDP architecture, the action $a$ is mapped to 256-dimensional space using linear network $A$. In this section, we make this distinction clear and use $a$ to denote raw action input to the network and $A$ to denote a projection of action as input to network. The outputs of these two networks are concatenated and passed through a feedforward neural network $g$ that has two hidden layers of 512 units and a $d$-dimensional output. The output of the $g$ network is passed through a linear layer $w$. The final $d$-dimensional representations are spherically normalized.

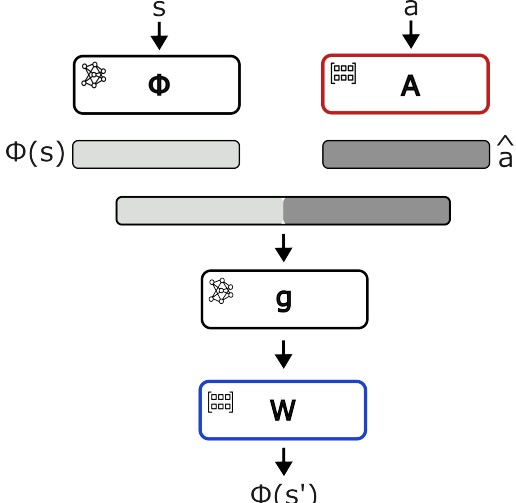

**Figure 6:** Architecture of latent next-state prediction network in RLDP.

*During encoder training*, the encoder map is unrolled to perform next latent state prediction from current latent state and action as $\phi(s') = g(\phi(s), A)^\top w$. *After encoder training*, the encoder network is frozen. To obtain latent state embeddings, the states are passed through the

state representation network to get $\phi(s)$. The encoder architecture for `RLDP` is kept consistent across all methods and datasets.

### A.4.2 ABLATIONS

In this section, we aim to examine the components of the `RLDP` state encoder to understand which parts of the method are crucial to learn representations that can maximize the span of reward functions we can represent optimal policies for.

We pretrain state representation network $\phi$ and policy using the ExoRL dataset generated with RND exploration policy and evaluate the performance in DMC environments cheetah, pointmass, quadruped, and walker.

**Does orthogonality regularization matter?**

Figure 7 shows the impact of changing orthogonality regularization while keeping a constant encoding horizon ($H = 5$). The figure shows how the cosine similarity between latent states changes during encoder training for different regularization coefficients.

For a regularization coefficient $\lambda = 0$, the cosine similarity increases, indicating that all states are getting mapped to similar representations. For any regularization coefficient $\lambda > 0$, we observe that the cosine similarity follows a steep descent, indicating that the states are being mapped to diverse representations. These results indicate that adding even small orthogonality regularization can reduce representation collapse significantly.

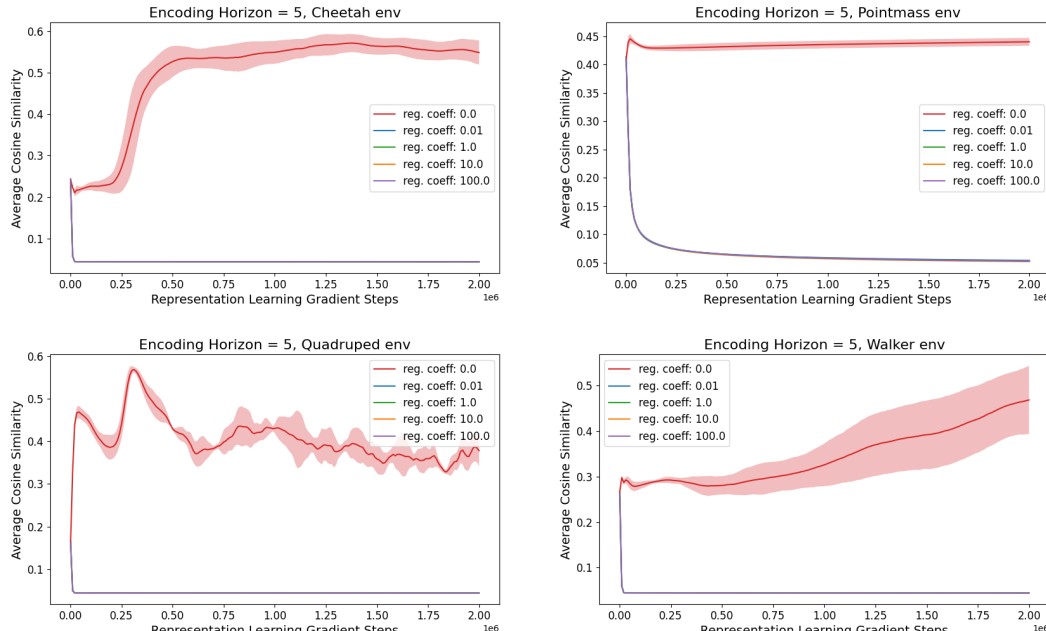

**Figure 7:** Evaluating the impact of Orthogonality Regularization on representations learned across four environments: Cheetah (top left), Pointmass (top right), Quadruped (bottom left), and Walker (bottom right).

**How does encoding horizon impact performance?** As discussed in section 4, `RLDP` is trained with the objective to do latent next state prediction from latent current state and action. This prediction can be done multiple steps into the future latent states (equation 6), depending on the choice of encoding horizon $H$.

In this section, we examine if the choice of encoding horizon impacts performance. To this end, we set the orthogonality regularization coefficient $\lambda = 1.0$ and sweep over encoding horizon $(1, 5, 10, 20)$.

The results are presented in figure 8. The average performance across environments is relatively stable with a small dip at $H = 10$, indicating that encoding horizon does not significantly impact performance. For our experiments, we use encoding horizon $H = 1$ or $H = 5$

depending on the setting. Specific encoding horizon setting for each experiment is discussed in section A.5. We do not choose higher encoding horizon $H = 20$ despite comparable performance in figure 8 because higher encoding horizon can result in slower encoder training. This is because each additional future state prediction involves a forward pass through the encoder network.

**What is important for the encoder architecture?**

In this section, we aim to ablate components of the encoder map to understand which factors contribute to `RLDP`'s performance. For this setting, we fix encoding horizon $H = 5$ and orthogonality regularization coefficient $\lambda = 1.0$.

We focus on two components of the encoder architecture – linear layer $A$ and spherical normalization $SN$ on $g$.

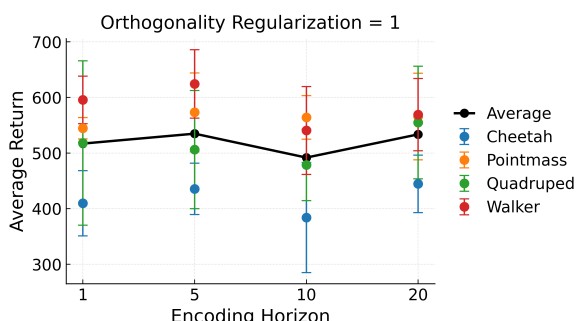

**Figure 8:** Evaluating the impact of Encoding Horizon

We compare the complete `RLDP` encoder network with its variations – a) *RLDP w/o SN* where spherical normalization is replaced with an identity mapping; b) *RLDP w/o A* where A is replaced with an identity mapping; c) *RLDP w/o SN & A* where A is also replaced with an identity mapping.

The results are shown in table 5. Although per-task results are variable, the full `RLDP` encoder delivered the strongest average performance on all four domains. Removing spherical normalization lowers returns and increases variance on most tasks and removing A also degrades performance. There are isolated wins for all variants, but these do not impact the domain-level results that favor the full `RLDP` encoder network. Thus, both $SN$ and $A$ contribute meaningfully to representation learning.

### A.4.3 WHAT DO THESE REPRESENTATIONS LOOK LIKE?

To qualitatively assess the learned state representations, we use the Pointmass environment, where we uniformly sampled 10,000 equidistant states from the underlying state space (figure 9 (a)).

We initialize a state representation encoder $\phi$ and pass these states through the encoder to get latent embedding before training (figure 9 (b)).

We then train two encoders with different losses: a. we set $\lambda = 0.0$ in equation 8 and train using only latent state prediction loss (figure 9 (c)); b. we set $\lambda = 1.0$ in equation 8 and train using latent state prediction loss and orthogonality regularization.

| | Task | RLDP | RLDP w/o SN | RLDP w/o A | RLDP w/o SN & A |
|---|---|---|---|---|---|
| **Walker** | Stand | **890.40±27.33** | 860.74±62.47 | 810.79±100.90 | 881.69±6.92 |
| | Run | **334.26±49.69** | 324.04±6.73 | 290.78±26.52 | 276.30±47.21 |
| | Walk | **779.77±137.16** | 728.29±43.09 | 715.83±92.43 | 583.60±28.26 |
| | Flip | 492.94±22.79 | **501.59±45.04** | 477.95±37.88 | 447.73±33.59 |
| | **Average(*)** | 624.34 | 603.66 | 573.84 | 547.33 |
| **Cheetah** | Run | **157.12±29.92** | 84.99±67.31 | 115.25±14.13 | 118.67±32.67 |
| | Run Backward | 170.52±15.30 | **193.69±40.10** | 192.20±42.07 | 156.56±45.98 |
| | Walk | **592.92±104.66** | 387.50±244.76 | 526.02±52.89 | 559.82±177.29 |
| | Walk Backward | 821.51±50.62 | **838.12±145.37** | 836.29±173.10 | 668.46±186.17 |
| | **Average(*)** | 435.52 | 376.08 | 417.44 | 375.88 |
| **Quadruped** | Stand | **794.94±43.25** | 661.73±95.75 | 518.61±69.24 | 687.43±155.33 |
| | Run | 457.41±74.70 | 378.97±148.47 | 358.55±53.61 | **475.07±45.66** |
| | Walk | 465.40±185.29 | 519.39±251.11 | 384.92±119.49 | **575.32±120.82** |
| | Jump | **733.32±55.30** | 495.98±133.81 | 319.18±55.16 | 510.55±151.18 |
| | **Average(*)** | 612.77 | 514.02 | 395.34 | 562.09 |
| **Pointmass** | Top Left | 890.41±60.79 | **892.13±41.74** | 886.19±10.07 | 890.89±13.06 |
| | Top Right | 795.47±21.10 | 728.72±122.99 | **809.64±11.23** | 797.59±19.44 |
| | Bottom Left | **805.17±20.44** | 683.12±76.22 | 730.74±63.72 | 735.42±61.83 |
| | Bottom Right | 193.38±167.63 | 22.54±39.04 | **206.59±214.98** | 178.77±130.17 |
| | **Average(*)** | 671.11 | 515.02 | 547.62 | 583.78 |

**Table 5:** Study of encoder architecture. Cells show mean ± std over 4 seeds; boldface indicates the highest mean per task.

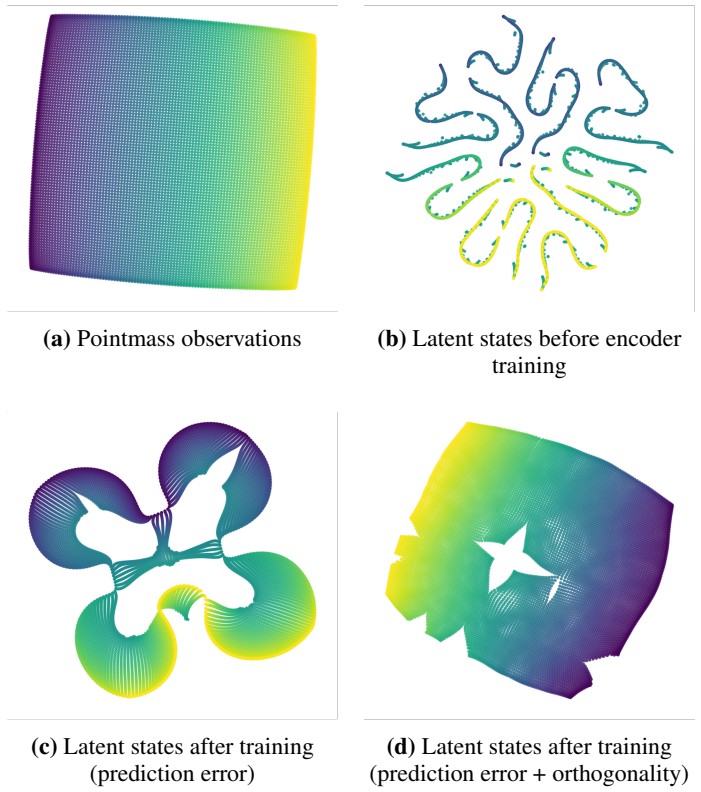

**(a)** Pointmass observations

**(b)** Latent states before encoder training

**(c)** Latent states after training (prediction error)

**(d)** Latent states after training (prediction error + orthogonality)

**Figure 9:** t-SNE visualizations of state features in Pointmass. Each panel shows the 2D projection of 10,000 uniformly sampled states.

We project all these embeddings into two dimensions t-distributed Stochastic Neighbor Embedding (t-SNE). This visualization highlights the geometric structure captured by the representation and provides intuition about how the encoder organizes states in latent space.

The results in figure 9 show that training an encoder using only latent state prediction loss (figure (c)) is ineffective at capturing the layout of the environment and maps different states to similar latent representations. Using both latent state prediction loss and orthogonality regularization enables the encoder to better capture the environment layout (figure(d)).

### A.4.4 FUTURE DIRECTIONS: EXTENSION TO REAL-WORLD EMBODIMENTS

`RLDP` presents a simple, stable and performant approach to train behavior foundation models in applications like robotics. An agent can be promptable to obtain low-level actions with such a BFM. Recent works (Tessler et al., 2025; Li et al., 2025) have made promising attempts to extend BFM algorithms to real-world domains and prior works have made it possible to prompt BFMs with language and videos (Sikchi et al., 2024) which can be more intuitive interface for humans than reward functions. We believe the simplicity of this method and stability across hyperparameter choices, as demonstrated in table 5, makes it a promising candidate for real-world embodiments.

### A.5 IMPLEMENTATION DETAILS

In this section, we discuss the implementation details of all the methods and experiments described in the paper.

### A.5.1 OFFLINE ZERO-SHOT RL

We use the same architecture for forward and policy networks as presented in (Touati et al., 2022) for all representation learning methods.

The forward network $F(s, a, z)$ has two parallel embedding layers that take in $(s, a)$ and $(s, z)$ independently using feedforward networks with a single hidden layer of 1024 units, projecting to 512 dimensions. Their outputs are concatenated and passed into two separate feedforward heads (each with one hidden layer of 1024 units), which output a $d$-dimensional vector.

The policy network $\pi(s, z)$ has two parallel embedding layers that take inputs $s$ and $(s, z)$ and embeds them similar to the forward network (one hidden layer of 1024 units mapping to 512 dimensions). The outputs of the embedding layers are concatenated, and then passed into another single-hidden-layer feedforward network (1024 units) to produce an action vector of dimension $d_A$. A final Tanh activation ensures that actions lie in the space $[-1, 1]^{d_A}$.

**For results in table 1:**

For all methods, the backward representation network $B(s)$ is implemented as a feedforward neural network with two hidden layers of 512 units each, mapping a state $s$ to a 512-dimensional embedding.

**For results in table 6:**

For $RLDP$, we sweep over representation dimensions $(64, 128, 256, 512, 1024)$ and report the results for the dimension that achieves the highest average performance across all tasks within each environment.

For *FB*, *PSM*, *HILP*, and *Laplacian*, we use the representation dimensions previously identified as optimal for each respective method.

### A.5.2    Online Zero-Shot RL

Results for oracle baseline, FB-CPR, are taken from Tirinzoni et al. (2025), where the model was trained for 30M environment steps and averaged across five seeds.

For the offline representation learning methods (HILP, PSM, FB, RLDP), the backward representation network $B(s)$ follows the architecture of the backward network of FB-CPR. It is a 2-layer MLP with 256 hidden dimension that maps a state $s$ to a 256-dimensional embedding. We train this for 2 million timesteps on a dataset provided by Tirinzoni et al. (2025), which is generated by online training an FB-CPR agent for 30 million environment steps and saving the final 5 million steps. The RLDP representations are trained with encoding horizon 1.

We integrate the learned representation network into an FB-CPR agent to train the forward and policy networks. This training is performed online for 20 million environment steps where no updates are performed on the representation network.

### A.5.3    Low Coverage Datasets

For all offline representation learning methods (HILP, PSM, FB, RLDP), the backward representation network $B(s)$ is a feedforward neural network with two hidden layers of 256 dimension that maps a state $s$ to a 512-dimensional embedding. The RLDP representations are trained with encoding horizon 1.

The forward network $F(s, a, z)$ and policy network $\pi(s, z)$ follow the same architecture as FB. We introduce an additional loss term for training the policy network that resembles TD3+BC (Fujimoto & Gu, 2021). The policy improvement loss is defined as

$$\mathcal{L}_P(\pi_z) = -\lambda \psi(s, a, z)^\top z + \left(\pi_z(s) - a\right)^2 \tag{15}$$

where

$$\lambda = \frac{\alpha}{\frac{1}{N} \sum_{(s_i, a_i)} \left|Q(s_i, a_i)\right|}.$$

Following Fujimoto & Gu (2021), we set $\alpha = 2.5$

## A.6 Algorithm

In algorithm 1, we present the full algorithm for pre-training and inference of BFMs. In pretraining, we present RLDP representation learning as well as successor-measure estimation and policy learning.

---

**Algorithm 1**

---

**Require:** Offline dataset of trajectories $\mathcal{D}$.
**Require:** Randomly initialized encoder $\phi$, successor-measure model $\psi$, actor $\pi$.
**Require:** Representation-learning steps $N_{\mathrm{repr}}$, total steps $N$.
1: **Part I: Pretraining (offline)**
2: **for** learning step $n = 1, 2, \ldots, N$ **do**
3:     **if** $n \leq N_{\mathrm{repr}}$ **then**
4:         Sample segment batch $\tau = \{s_{0:H}^i, a_{0:H}^i\}_{i=1}^B \sim \mathcal{D}$
5:         $h_0^i = \phi(s_0^i), \quad h_{t+1}^i = g(h_t^i, a_t^i)^\top \mathbf{w}$
6:         $\mathcal{L}_d(\phi, g, w) = \mathbb{E}_{\tau^i \sim d^O}\left[\left\|\sum_{t=1}^H h_t^i - \bar{\phi}(s_t^i)\right\|^2\right]$
7:         $\mathcal{L}_r(\phi) = \mathbb{E}_{s,s' \sim \rho}\left[\phi(s)^\top \phi(s')\right]$
8:         $L(\phi, g, w) \leftarrow \mathcal{L}_d(\phi, g, w) + \lambda \mathcal{L}_r(\phi)$
9:         Update $\phi, g, w$
10:     **else**
11:         Sample transitions $\{(s, a, s', \mathrm{done})\} \sim \mathcal{D}$
12:         Sample $z \sim$ Uniform Mix{random prior + goal-encoded}
13:         **Policy Evaluation**:
14:         $L_{zsrl}(\psi)$ from Equation 10
15:         $\psi \leftarrow \psi - \alpha_\psi L_{zsrl}(\psi)$
16:         **Policy update**:
17:         $a \sim \pi(s, z)$
18:         $Q = \psi(s, a, z) \cdot z$
19:         $\pi \leftarrow \pi + \alpha_\pi \nabla_\pi Q(s, \pi(s, z))$
20:     **end if**
21: **end for**
22:
23: **Part II: Inference (reward-based task embedding)**
**Require:** Task specification for the test task (e.g., name, parameters).
24: Set up the task-specific reward function $r_{\mathrm{task}}(s)$ using the environment's reward routine
25: Sample transitions $\{(s_i, a_i, s_i')\}_{i=1}^N \sim \mathcal{D}$
26: $z \leftarrow \dfrac{1}{N} \sum_i \phi(s_i') r_{task}(s_i')$

---

## A.7 Additional results

This section details additional experiments we conducted to evaluate RLDP against baseline methods, visualize the successor measures learned by RLDP, and study the effect of different loss components on representation learning.

### A.7.1 Offline Zero-shot RL in DMC

In table 6, we compare the returns for the representation dimension found to be the best for the baseline methods and the RLDP. Across all methods, RLDP fares competitively to baselines that employ complex strategies such as FB, PSM to learn representation optimizing for successor measures across the environments despite its simplicity.

| | Task | Laplace | FB | HILP | PSM | RLDP |
|---|---|---|---|---|---|---|
| **Walker** | Stand | 243.70±151.40 | **902.63±38.94** | 607.07±165.28 | **872.61±38.81** | **877.69±45.03** |
| | Run | 63.65±31.02 | **392.76±31.29** | 107.84±34.24 | 351.50±19.46 | 324.85±54.57 |
| | Walk | 190.53±168.45 | **877.10±81.05** | 399.67±39.31 | **891.44±46.81** | 790.94±67.55 |
| | Flip | 48.73±17.66 | 206.22±162.27 | 277.95±59.63 | **640.75±31.88** | 491.64±37.30 |
| **Cheetah** | Run | 96.32±35.69 | **257.59±58.51** | 68.22±47.08 | **244.38±80.00** | **236.31±20.75** |
| | Run Backward | 106.38±29.40 | **307.07±14.91** | 37.99±25.16 | **296.44±20.14** | 322.08±39.28 |
| | Walk | 409.15±56.08 | 799.83±67.51 | 318.30±168.42 | **984.21±0.49** | 895.31±49.84 |
| | Walk Backward | 654.29±219.81 | **980.76±2.32** | 349.61±236.29 | **979.01±7.73** | **984.76±0.85** |
| **Quadruped** | Stand | **854.50±41.47** | 740.05±107.15 | 409.54±97.59 | **842.86±82.18** | 794.94±43.25 |
| | Run | **412.98±54.03** | 386.67±32.53 | 205.44±47.89 | 431.77±44.69 | 457.41±74.70 |
| | Walk | 494.56±62.49 | **566.57±53.22** | 218.54±86.67 | **603.97±73.67** | 465.40±185.29 |
| | Jump | 642.84±114.15 | 581.28±107.38 | 325.51±93.06 | 596.37±94.23 | **733.32±55.30** |
| **Pointmass** | Top Left | 713.46±58.90 | 897.83±35.79 | **944.46±12.94** | 831.43±69.51 | 890.41±60.79 |
| | Top Right | 581.14±214.79 | 274.95±197.90 | 96.04±166.34 | 730.27±58.10 | **795.47±21.10** |
| | Bottom Left | 689.05±37.08 | 517.23±302.63 | 192.34±177.48 | 451.38±73.46 | **805.17±20.44** |
| | Bottom Right | 21.29±42.54 | 19.37±33.54 | 0.17±0.29 | **43.29±38.40** | **193.38±167.63** |

**Table 6:** Comparison of zero-shot offline RL performance between different methods. Entries in **bold** are within one standard deviation of the per-task best mean (i.e., $\mu_i \geq \mu^* - \sigma^*$) aggregated over 4 seeds.

## A.8 TRAINING USFAS ON TOP OF RLDP REPRESENTATIONS

In table 7, we examine the impact of directly learning Universal Successor Features on top of RLDP representations. Typically for offline zero-shot RL on RLDP representations, we use loss equation 10 to update the critic network. To train USFAs, we use the following loss:

$$\mathcal{L}_{USFA}(\psi) = \mathbb{E}_{s,a,s'\sim\rho,s^+\sim\rho}[(\psi(s,a,z) - [\phi(s) + \gamma\bar{\psi}(s', \pi_z(s'), z)])^2] \qquad (16)$$

We find that across a wide range of control tasks, training a USFA module on top of RLDP's state representations does not consistently outperform directly using successor measure loss 10 for policy evaluation. Critic learned using successor measure loss achieves overlapping-best performance on most Walker, Quadruped, and Pointmass tasks, while USFAs occasionally match or slightly exceed on certain Cheetah behaviors. Overall, these results indicate that RLDP's learned representations capture most of the structure required for effective zero-shot generalization using either loss. The critic trained with successor measure loss typically achieves the strongest overall performance.

| | Task | RLDP | Learning USFAs on RLDP representations |
|---|---|---|---|
| **Walker** | Stand | 890.40±27.33 | 854.84±64.50 |
| | Run | 334.26±49.69 | 304.89±63.15 |
| | Walk | 779.77±137.16 | 665.98±117.64 |
| | Flip | 492.94±22.79 | 497.77±70.80 |
| **Cheetah** | Run | 157.12±29.92 | 139.00±6.72 |
| | Run Backward | 170.52±15.30 | 172.52±25.72 |
| | Walk | 592.92±104.66 | 647.35±66.05 |
| | Walk Backward | 821.51±50.62 | 800.02±82.94 |
| **Quadruped** | Stand | 794.94±43.25 | 294.09±165.70 |
| | Run | 457.41±74.70 | 238.42±121.54 |
| | Walk | 465.40±185.29 | 326.36±111.55 |
| | Jump | 733.32±55.30 | 220.66±67.59 |
| **Pointmass** | Top Left | 890.41±60.79 | 723.80±55.33 |
| | Top Right | 795.47±21.10 | 723.79±57.89 |
| | Bottom Left | 805.17±20.44 | 753.66±16.59 |
| | Bottom Right | 193.38±167.63 | 94.39±77.81 |

**Table 7:** Comparison (over 4 seeds) of zero-shot RL performance when equation 10 is used to train the critic and when Universal Successor Features are trained on top of the state features. For fair comparison, we set RLDP representation dimension $d = 512$ for both methods.

### A.8.1 ONLINE ZERO-SHOT RL

FB-CPR is an off-policy online unsupervised RL algorithm that introduces a latent conditional-discriminator in the form of Conditional-Policy Regularization to output policies close to an unlabeled demonstration dataset $\mathcal{M}$. The results for FB-CPR are as reported in Tirinzoni et al. (2025).

In table 8 and figure 10, we provide the full suite of results on 45 SMPL Humanoid task for all baseline methods, RLDP, and the oracle method FB-CPR.

| metric | FB-CPR (Oracle) | FB | PSM | HILP | RLDP |
|---|---|---|---|---|---|
| crawl-0.4-0-d | 191.75 ± 43.60 | 26.06 ± 39.43 | 38.38 ± 14.73 | 52.31 ± 23.15 | **86.48 ± 45.87** |
| crawl-0.4-0-u | 101.76 ± 15.90 | 8.38 ± 9.01 | 4.52 ± 7.47 | 18.59 ± 20.42 | **25.00 ± 27.32** |
| crawl-0.4-2-d | 19.00 ± 4.00 | 3.05 ± 3.65 | 6.52 ± 1.27 | 8.97 ± 5.62 | **11.21 ± 4.72** |
| crawl-0.4-2-u | 15.02 ± 6.03 | 0.79 ± 1.07 | 0.64 ± 0.82 | **2.95 ± 1.37** | 2.76 ± 3.73 |
| crawl-0.5-0-d | 131.13 ± 64.97 | 43.27 ± 34.66 | 46.17 ± 13.56 | 52.41 ± 27.82 | **55.82 ± 18.40** |
| crawl-0.5-0-u | 101.92 ± 16.39 | 4.04 ± 5.87 | 4.18 ± 4.83 | **21.14 ± 24.26** | 20.22 ± 30.91 |
| crawl-0.5-2-d | 22.93 ± 5.31 | 4.14 ± 4.10 | 5.64 ± 1.79 | **8.64 ± 4.21** | 5.69 ± 2.18 |
| crawl-0.5-2-u | 15.81 ± 6.10 | 0.94 ± 0.99 | 0.77 ± 0.70 | 2.67 ± 1.03 | **2.95 ± 3.28** |
| crouch-0 | 226.28 ± 28.17 | 55.12 ± 47.09 | **92.70 ± 60.86** | 72.94 ± 76.25 | 4.83 ± 5.28 |
| headstand | 41.27 ± 10.20 | 0.00 ± 0.00 | 0.00 ± 0.01 | 0.11 ± 0.16 | **2.63 ± 1.99** |
| jump-2 | 34.88 ± 3.52 | **29.08 ± 3.76** | 21.21 ± 11.60 | 12.25 ± 14.05 | 27.89 ± 1.66 |
| lieonground-down | 193.50 ± 18.89 | 35.41 ± 26.08 | 63.87 ± 26.68 | 69.79 ± 27.02 | **74.69 ± 30.03** |
| lieonground-up | 193.66 ± 33.18 | 20.83 ± 12.70 | 13.92 ± 5.54 | 30.81 ± 1.37 | **54.38 ± 31.06** |
| move-ego–90-2 | 210.99 ± 6.55 | **207.47 ± 9.92** | 179.67 ± 49.64 | 196.81 ± 40.36 | 178.82 ± 45.10 |
| move-ego–90-4 | 202.99 ± 9.33 | **161.84 ± 12.65** | 102.35 ± 35.15 | 102.98 ± 40.47 | 99.96 ± 32.58 |
| move-ego-0-0 | 274.68 ± 1.48 | 261.63 ± 1.76 | 264.32 ± 1.95 | **267.57 ± 0.95** | 178.92 ± 92.57 |
| move-ego-0-2 | 260.93 ± 5.21 | 87.46 ± 21.99 | 252.75 ± 15.03 | **260.35 ± 2.58** | 250.92 ± 6.75 |
| move-ego-0-4 | 235.44 ± 29.42 | 133.47 ± 33.86 | **234.14 ± 8.81** | 233.02 ± 14.75 | 201.90 ± 38.55 |
| move-ego-180-2 | 227.34 ± 27.01 | **232.14 ± 20.35** | 141.56 ± 32.41 | 139.12 ± 83.74 | 222.83 ± 28.29 |
| move-ego-180-4 | 205.54 ± 14.40 | **109.04 ± 27.89** | 71.42 ± 19.98 | 53.37 ± 25.65 | 81.92 ± 29.90 |
| move-ego-90-2 | 210.99 ± 6.55 | 217.16 ± 26.35 | 214.64 ± 37.08 | 178.96 ± 45.28 | **221.43 ± 33.90** |
| move-ego-90-4 | 202.99 ± 9.33 | 154.20 ± 41.82 | 104.73 ± 20.95 | 102.51 ± 63.37 | **160.31 ± 37.02** |
| move-ego-low–90-2 | 221.37 ± 35.35 | 75.28 ± 29.80 | 76.96 ± 49.83 | **126.80 ± 80.76** | 30.15 ± 26.04 |
| move-ego-low-0-0 | 215.61 ± 27.63 | 168.33 ± 5.95 | 150.34 ± 62.07 | **188.29 ± 49.41** | 133.68 ± 52.10 |
| move-ego-low-0-2 | 207.27 ± 58.01 | 82.66 ± 20.55 | 73.60 ± 49.86 | **104.77 ± 23.00** | 66.84 ± 44.92 |
| move-ego-low-180-2 | 65.20 ± 32.64 | **52.38 ± 27.67** | 46.28 ± 22.28 | 43.90 ± 39.86 | 28.71 ± 12.52 |
| move-ego-low-90-2 | 222.81 ± 21.94 | **100.75 ± 39.27** | 53.20 ± 21.26 | 85.54 ± 82.04 | 63.19 ± 42.99 |
| raisearms-h-h | 199.88 ± 42.03 | 192.49 ± 101.91 | 94.64 ± 94.26 | 171.41 ± 71.90 | **217.09 ± 34.35** |
| raisearms-h-l | 167.98 ± 82.03 | **226.33 ± 35.55** | 90.57 ± 68.37 | 82.42 ± 43.38 | 201.33 ± 87.51 |
| raisearms-h-m | 104.26 ± 81.69 | 100.49 ± 76.12 | 61.82 ± 20.38 | 112.16 ± 76.75 | **155.36 ± 85.85** |
| raisearms-l-h | 243.16 ± 19.18 | **255.41 ± 1.55** | 128.56 ± 63.06 | 136.49 ± 85.25 | 233.82 ± 27.26 |
| raisearms-l-l | 270.43 ± 0.37 | 251.82 ± 9.70 | **260.48 ± 3.52** | 258.50 ± 6.07 | 39.87 ± 34.16 |
| raisearms-l-m | 97.66 ± 81.17 | 135.05 ± 80.31 | **254.91 ± 3.78** | 91.49 ± 46.58 | 217.42 ± 39.30 |
| raisearms-m-h | 75.05 ± 69.32 | 79.25 ± 31.99 | 41.58 ± 13.58 | **126.62 ± 80.07** | 107.70 ± 79.18 |
| raisearms-m-l | 134.83 ± 70.28 | 218.22 ± 46.82 | 173.28 ± 72.83 | 155.21 ± 71.93 | **220.67 ± 50.89** |
| raisearms-m-m | 87.25 ± 98.42 | 179.60 ± 74.63 | 109.47 ± 91.62 | 82.36 ± 38.59 | **211.30 ± 48.89** |
| rotate-x–5-0.8 | 2.29 ± 1.78 | 1.69 ± 2.32 | 1.49 ± 1.43 | 0.29 ± 0.18 | **2.44 ± 2.02** |
| rotate-x-5-0.8 | 7.42 ± 5.69 | 2.55 ± 1.29 | 0.53 ± 0.43 | 0.32 ± 0.28 | **6.43 ± 3.15** |
| rotate-y–5-0.8 | 199.08 ± 51.78 | 5.87 ± 3.63 | 2.13 ± 2.17 | 1.04 ± 0.11 | **8.18 ± 4.71** |
| rotate-y-5-0.8 | 217.70 ± 43.67 | 4.86 ± 1.44 | 1.58 ± 0.44 | 0.89 ± 0.13 | **14.03 ± 12.12** |
| rotate-z–5-0.8 | 124.95 ± 17.61 | 0.72 ± 0.79 | 0.42 ± 0.30 | 0.31 ± 0.23 | **17.09 ± 9.10** |
| rotate-z-5-0.8 | 95.23 ± 15.75 | **1.71 ± 1.67** | 0.39 ± 0.37 | 0.38 ± 0.22 | 0.66 ± 0.76 |
| sitonground | 199.44 ± 22.15 | 5.88 ± 4.75 | 27.39 ± 22.19 | 26.12 ± 21.69 | **97.88 ± 34.91** |
| split-0.5 | 232.18 ± 20.26 | 12.64 ± 14.48 | 34.31 ± 32.98 | **87.22 ± 5.92** | 55.50 ± 33.46 |
| split-1 | 117.67 ± 61.27 | 6.80 ± 9.14 | 6.12 ± 7.17 | 6.13 ± 5.72 | **13.02 ± 16.90** |

**Table 8:** Comparing (over 4 seeds) FB, PSM, HILP, RLDP performance on SMPL Humanoid. FB-CPR (online oracle baseline) results are from Tirinzoni et al. (2025). Bold indicates the best mean across methods.

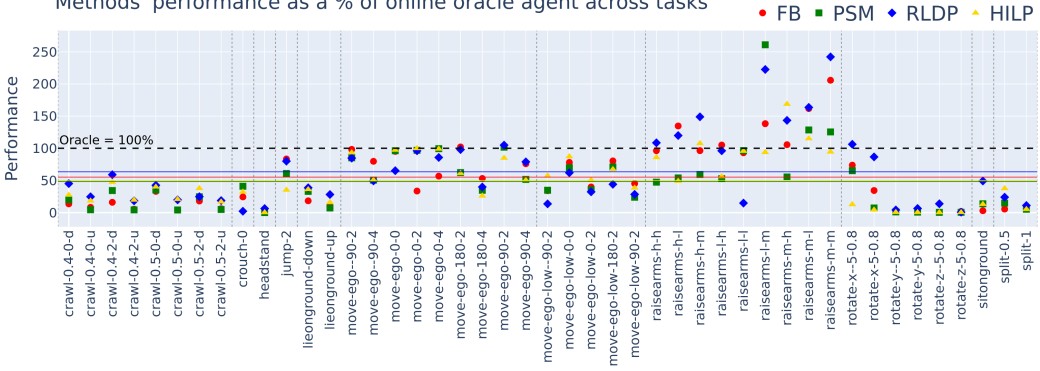

**Figure 10:** Evaluating offline representation learning methods using an online oracle policy in high dimensional 3D humanoid. Solid lines shows mean performance across tasks for each method.

## A.9 FULL RESULTS ON D4RL

Table 9 shows the normalized average returns of RLDP and baseline methods on six low-coverage D4RL environments over 10 seeds. RLDP achieves the best mean performance on 5/6 tasks. Using Welch's t-tests with Holm correction, RLDP significantly outperforms FB on all tasks and

| Task | FB | PSM | HILP | RLDP |
|---|---|---|---|---|
| halfcheetah-medium-expert-v2 | 52.17±11.37 | 49.92±26.89 | 68.47±8.20 | **86.03±8.36** |
| halfcheetah-medium-v2 | 39.27±8.71 | 42.64±0.64 | 43.85±1.49 | **49.08±1.93** |
| hopper-medium-expert-v2 | 54.64±19.47 | 14.59±26.35 | 68.18±18.19 | **77.21±16.77** |
| hopper-medium-v2 | 43.75±6.65 | 33.49±26.71 | **52.19±4.02** | 44.93±13.08 |
| walker2d-medium-expert-v2 | 60.17±28.94 | 79.32±41.16 | 93.72±21.12 | **103.87±3.31** |
| walker2d-medium-v2 | 43.72±24.95 | 55.70±22.01 | 56.34±15.08 | **83.83±2.66** |

**Table 9:** Normalized returns comparing FB, PSM, HILP, and RLDP in low-coverage setting. RLDP shows significant gains over approaches that rely on explicit Bellman backups for representation learning. Table shows mean±std over 10 seeds; **Boldface** indicates the highest mean return per environment. Statistical comparisons use per-seed returns with Welch's t-test and Holm correction; cases where the bolded method is not significantly better than the runner-up are discussed in the text..

significantly outperforms PSM on 5/6 tasks. RLDP's gains over HILP are statistically significant on 3/6 tasks, while differences on Hopper-medium-expert and Walker2d-medium-expert are not statistically decisive with 10 seeds due to high variance. On Hopper-medium, HILP has the highest mean but the HILP–RLDP difference is not significant under Welch's test, indicating comparable performance under seed variability.

## A.10 VISUALIZATIONS OF LEARNED SUCCESSOR MEASURES

We used a four room gridworld (as used in Touati & Ollivier (2021); Agarwal et al. (2024)) to plot the successor measures learned by RLDP. We collect a dataset of all transitions and run RLDP with horizon 1 to learn representations $\phi$, successor features $\psi$ and policy $\pi$. Note that any policy parameterized by latent $z$ produces a successor measure parameterized by $M^{\pi_z}(s, a, s^+) = \psi(s, a, z)^T \phi(s^+)$. We have plotted the observed successor measures: $M^{\pi_z}(s_0, a_0, s^+)$, where we fix $s_0$ and $a_0$ for a few different $z$ in figure 11. We have fixed $s_0$ to the corner state and $a_0$ to action: right. We have plotted the policy for visualizing the policy represent by $z$.

## A.11 EVALUATING THE EFFECT OF DIFFERENT LOSSES ON REPRESENTATION LEARNING

In table 10, we ablate the objectives used to learn representations and compare (a) Bellman-style loss used in FB, (b) latent prediction loss used in RLDP, and c) orthogonality regularization, that both FB and RLDP use. We evaluate these loss objectives individually and combined with unit scaling.

$$\mathcal{L}_{\text{Bellman}}(\phi, \psi) = -\mathbb{E}_{s,a,s'\sim\rho}[\psi(s, a, z)^T \phi(s')]$$
$$+ \frac{1}{2}\mathbb{E}_{s,a,s'\sim\rho,s^+\sim\rho}[(\psi(s, a, z)^T \phi(s^+) - \gamma\bar{\psi}(s', \pi_z(s'), z)^T \bar{\phi}(s^+))^2] \quad (17)$$

$$\mathcal{L}_{\text{Ortho}}(\phi) = \mathbb{E}_{s,s'\sim\rho}[\phi(s)^\top \phi(s')] \quad (18)$$

$$\mathcal{L}_{\text{Prediction}}(\phi, g, w) = \mathbb{E}_{\tau^i\sim d^O}\left[\sum_{t=0}^{H-1} \left\|h_{t+1}^i - \bar{\phi}(s_{t+1}^i)\right\|_2^2\right], \qquad h_0^i = \phi(s_0^i), h_{t+1}^i = g(h_t^i, a_t^i)\, w \quad (19)$$

The results in table 10 highlight that the interaction between different objectives matters much more than any individual loss in isolation. Across all three expert datasets, we find that a combined objective of prediction loss and orthogonality regularization yield the largest returns. In addition, adding Bellman backup to this objective with unit scaling results in inconsistent final returns.

Overall, these results support the design choice of using latent prediction with orthogonal regularization as the primary representation learning objective in low-coverage settings due to its high-performance and robustness across multiple settings.

We leave a more exhaustive study of this setting to future work, including evaluating these objectives on a wider set of environments and exploring principled scaling for the loss components.

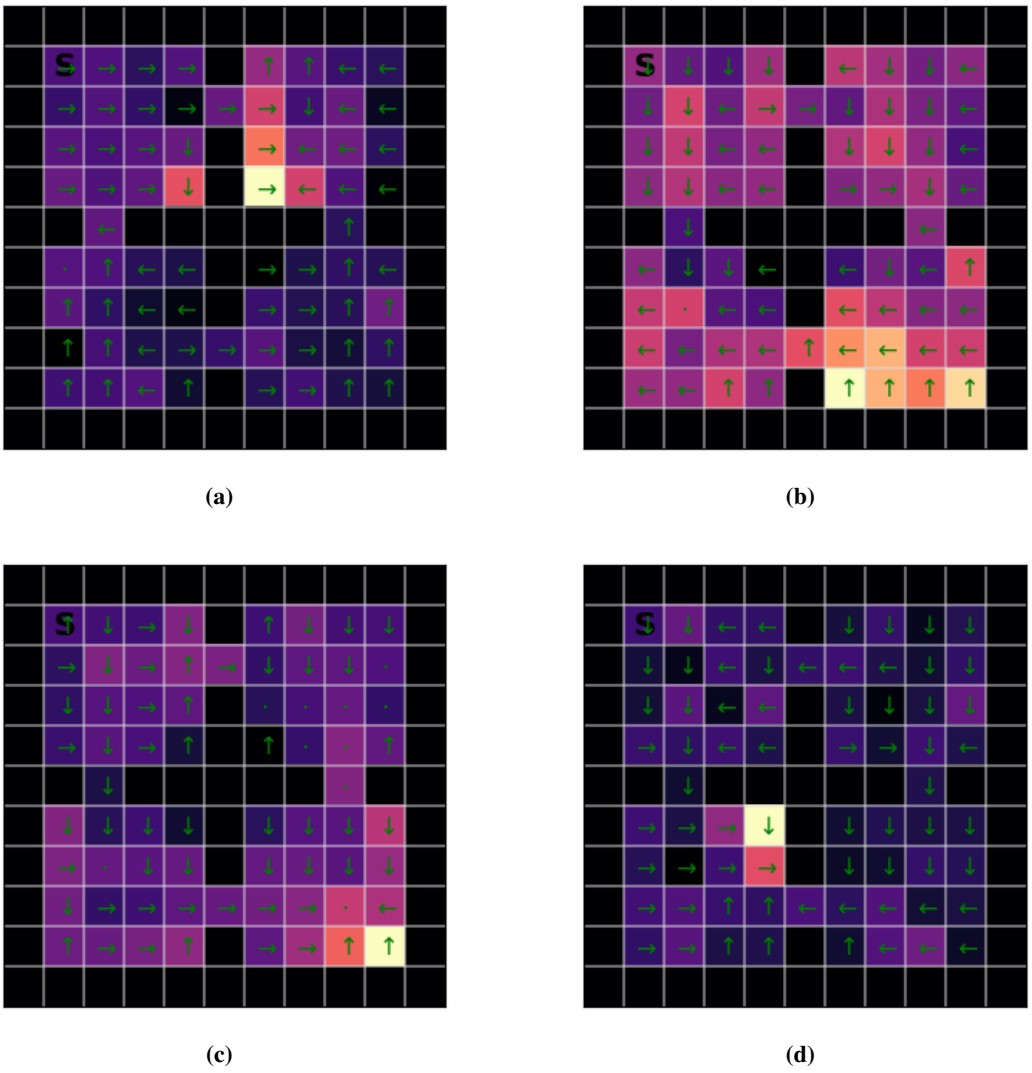

**(a)**         **(b)**

**(c)**         **(d)**

**Figure 11:** Visualization of successor measures $M^{\pi_z}(s_0, a_0, s^+)$ for randomly sampled $z$ (a) and (b); and goal-conditioned $z$ (c) and (d).

| Environment | Returns (mean $\pm$ std) |
|---|---|
| `halfcheetah-medium-expert` | |
| $\mathcal{L}_{\text{Ortho}}$ | $51.36 \pm 6.93$ |
| $\mathcal{L}_{\text{Bellman}}$ | $64.42 \pm 7.59$ |
| $\mathcal{L}_{\text{Prediction}}$ | $49.00 \pm 4.34$ |
| $\mathcal{L}_{\text{Bellman}} + \mathcal{L}_{\text{Prediction}} + \mathcal{L}_{\text{Ortho}}$ | $89.45 \pm 3.81$ |
| $\mathcal{L}_{\text{Bellman}} + \mathcal{L}_{\text{Ortho}}$ (**FB**) | $55.46 \pm 7.75$ |
| $\mathcal{L}_{\text{Prediction}} + \mathcal{L}_{\text{Ortho}}$ (`RLDP`) | $88.55 \pm 6.31$ |
| `hopper-medium-expert` | |
| $\mathcal{L}_{\text{Ortho}}$ | $56.13 \pm 3.40$ |
| $\mathcal{L}_{\text{Bellman}}$ | $59.02 \pm 12.19$ |
| $\mathcal{L}_{\text{Prediction}}$ | $54.63 \pm 3.51$ |
| $\mathcal{L}_{\text{Bellman}} + \mathcal{L}_{\text{Prediction}} + \mathcal{L}_{\text{Ortho}}$ | $49.25 \pm 11.90$ |
| $\mathcal{L}_{\text{Bellman}} + \mathcal{L}_{\text{Ortho}}$ (**FB**) | $49.93 \pm 28.78$ |
| $\mathcal{L}_{\text{Prediction}} + \mathcal{L}_{\text{Ortho}}$ (`RLDP`) | $75.53 \pm 12.70$ |
| `walker2d-medium-expert` | |
| $\mathcal{L}_{\text{Ortho}}$ | $95.10 \pm 12.28$ |
| $\mathcal{L}_{\text{Bellman}}$ | $89.12 \pm 19.36$ |
| $\mathcal{L}_{\text{Prediction}}$ | $97.52 \pm 14.27$ |
| $\mathcal{L}_{\text{Bellman}} + \mathcal{L}_{\text{Prediction}} + \mathcal{L}_{\text{Ortho}}$ | $27.10 \pm 22.51$ |
| $\mathcal{L}_{\text{Bellman}} + \mathcal{L}_{\text{Ortho}}$ (**FB**) | $36.85 \pm 14.89$ |
| $\mathcal{L}_{\text{Prediction}} + \mathcal{L}_{\text{Ortho}}$ (`RLDP`) | $101.30 \pm 3.92$ |

**Table 10:** Results in D4RL expert environments across different loss combinations aggregated over 4 seeds.

