# OpenReview forum: "Regularized Latent Dynamics Prediction is a Strong Baseline For Behavioral Foundation Models"
_ICLR.cc/2026/Conference — ICLR 2026 Poster_

### Official Review · Reviewer_GfJN · 2025-11-01

**Soundness:** 3
**Presentation:** 3
**Contribution:** 2
**Rating:** 6
**Confidence:** 4

**Summary:**

This paper proposes to use regularized latent dynamics prediction as an objective to learn representations for behavior foundation models. The main idea is to learn a state representation using a latent dynamics prediction objective, where the encoded latent states $\phi(s_t)$ are trained to regress the unrolled predicted latent states $h_t := g(h_{t-1}), a_t, h_0 = \phi(s_0)$. This objective alone suffers from collapse (shown in Figure 1), so the paper proposes to use an orthogonal regularization to mitigate it. The state representation is then used as the backward feature in the Forward-Backward (FB) representation framework. The paper shows theoretically that optimizing the latent dynamics objective also reduces prediction error in successor measures (i.e. the FB objective). Empirically, they evaluate the method on the offline zero-shot RL benchmark ExoRL, where the method matches or outperforms other behavior foundation model baselines. Their method also achieves online zero-shot RL in a Humanoid environment and is capable of learning representations from low-coverage datasets. The key design choice of orthogonal regularization is justified by ablation experiments. Overall, the paper proposes a simple yet effective method for zero-shot RL.

**Strengths:**

1. The proposed method is simple yet effective, performing favorably to baselines across multiple settings while suffering less from feature collapse.
2. The theoretical justification is sound.
3. The paper conducts thorough analysis on the feature collapse phenomenon and the effect of orthonogal regularization.

**Weaknesses:**

1. The theory and practice are somewhat mismatched. Theory says that learning successor measure on top of the latent representations improves as the representation learning objective improves. However, in practice, the forward feature is learned from scratch, and only the backward feature is carried over.
2. The method does not improve much compared to baselines in the ExoRL benchmark.

**Questions:**

1. Can you explain the mismatch between theory and practice as outlined in Weakness 1?
2. Can you try an experiment where you learn a Universal Successor Feature on top of the state features, and do zero-shot RL with linear regression (assuming the features linearly span rewards)? This can potentially substantiate the broad applicability of the learned representations.

References:

[1] Diana Borsa, André Barreto, John Quan, Daniel Mankowitz, Rémi Munos, Hado van Hasselt, David Silver, Tom Schaul. Universal Successor Features Approximators. ICLR 2019.

---

> ### Author Response · Authors · 2025-11-24
> **Response to Reviewer GfJN**
>
> We thank the reviewer for their feedback on our work and encouraging comments. We are happy to learn that reviewer appreciated the simplicity of the method, along with sound theoretical justification and thorough analysis. We address the questions below:
>
> >  Theory says that learning successor measure on top of the latent representations improves as the representation learning objective improves. However, in practice, the forward feature is learned from scratch, and only the backward feature is carried over.
>
> We note that RLDP learns representations independent of the successor measure and policy learning. For the representation learning phase, we train one encoder with loss in eq 8 which does not estimate successor measure, after which we freeze the encoder to train the forward feature and policy. The RLDP loss is a loss on dynamics prediction with the orthonormal regularization. While we show that $L_{RLDP}$ upper bounds the prediction error for successor measures, optimizing over only the dynamics is not sufficient to learn the functional form of successor measures that are needed for BFMs. We hope that the Algorithm in appendix A.6 adds clarity explaining the complete method. Lemma A.1. is intended to illustrate that if a successor measure is learned in MDP corresponding to state abstraction (assuming perfect successor measure learning) the prediction error would be bounded by RLDP loss.The loss for learning successor measures (in Equation 3) has successor measure as its fixed point [1] but in practice none of the methods actually succeed in minimizing the loss completely, meaning the representation $\psi(s, a, z) ^T \phi(s^+)$ do not accurately represent successor measures upon empirical convergence. We motivate the representation learning objective of $\phi$ to explain that the successor measure learning for the zero-shot RL can be easier.
>
>
> > The method does not improve much compared to baselines in the ExoRL benchmark.
>
> RLDP is competitive to baselines in the ExoRL benchmark. We found the ExoRL benchmark to be saturated to some extent but it serves as an important benchmark regardless because it validates that our simple approach can be competitive. We compare an additional 45 tasks in SMPL-based Humanoid environment (a realistic simulator) and 6 tasks from D4RL to show the merits of RLDP.
>
>
> > Can you try an experiment where you learn a Universal Successor Feature on top of the state features?
>
> Thanks for suggesting this experiment. The key difference is that USF estimates successor features whereas RLDP estimates the successor measure. We add an ablation in Table 8 where we see that USF also enables zero shot RL albeit is less performant than learning successor measures.
>
> [1]: Blier et. al, Learning Successor States and Goal Dependent values: A Mathematical Viewpoint, 2021
>
> ---------------
>
>
> Please let us know if you have any further questions!

---

### Official Review · Reviewer_Th7h · 2025-11-01

**Soundness:** 4
**Presentation:** 3
**Contribution:** 3
**Rating:** 6
**Confidence:** 2

**Summary:**

This work presents Regularized Latent Dynamics Prediction (RLDP), a representation learning approach for zero-shot reinforcement learning that replaces complex, reward-dependent objectives with a simple, self-supervised latent dynamics prediction task. The method augments next-state prediction with an orthogonality regularization term that mitigates feature similarity and prevents collapse in the learned state representations. Empirical evaluations across multiple benchmarks—including DeepMind Control Suite, D4RL, and high-dimensional humanoid control—show that RLDP achieves performance comparable to or exceeding state-of-the-art Behavioral Foundation Model baselines such as FB, PSM, and HILP. In particular, RLDP maintains stable performance in low-coverage datasets.

**Strengths:**

The paper makes a simple yet effective case for introducing latent dynamics into zero-shot offline RL.

RLDP achieves performance on par with or exceeding that of more complex methods, demonstrating that simple latent dynamics prediction can yield strong generalization.

Policy-independent formulation avoids instability from Bellman backups, leading to more reliable learning in challenging or low-coverage environments.

**Weaknesses:**

Evaluation is performed on simulated continuous control benchmarks. Real-world data or transfer is not evaluated.

Empirical results show comparable but not universally superior performance, suggesting that the method’s advantages depend on task characteristics and data diversity.

**Questions:**

What challenges do the authors suspect when adapting RLDP to real-world domains?

How would the approach perform in settings where the dynamics themselves evolve or where the environment exhibits non-stationarity—common in human-in-the-loop or adaptive control scenarios?

---

> ### Author Response · Authors · 2025-11-24
> **Response to Reviewer Th7h**
>
> We thank the reviewer for their evaluation of our work. We are encouraged that the reviewer found our work simple yet effective and appreciated the reliability of the method as a result of policy independent formulation.
>
> > What challenges do the authors suspect when adapting RLDP to real-world domains?
>
> We believe RLDP is a promising candidate for real-world domains. One of key reasons is that we are less sensitive to hyperparameters, making it so that RLDP should be comparatively easier to tune (Tables 3 and 6 shows the stability across hyperparameter choices). RLDP shows stable training across a wide variety of domains which are more diverse than evaluated in prior works [1,2,3] including a humanoid modelled after real-human dynamics [SMPL, 4]. There is nothing conceptually preventing application of RLDP to real-world domains, though we suspect the typical challenges when using RL in the real-world will arise (e.g., picking hyperparameters with minimal environment interaction). We have added a discussion in appendix section A.4.4.
>
>
> > How would the approach perform in settings where the dynamics themselves evolve or where the environment exhibits non-stationarity—common in human-in-the-loop or adaptive control scenarios?
>
> A natural future extension of our work here is to use offline-to-online RL (also called hybrid RL). The policy can continue to adapt with new data. RLDP is not restricted to using only a fixed policy, and can be layered on-top of many different base offline algorithms or offline-to-online RL algorithms. RLDP primarily learns a representation that facilitates transfer. This representation itself could be slowly adapted online, and the base RL algorithm can also learn online.
>
>
> [1]: Agarwal, Siddhant, et al. "Proto Successor Measure: Representing the Behavior Space of an RL Agent." ICML 2025.
> [2]: Touati, Ahmed, Jérémy Rapin, and Yann Ollivier. "Does zero-shot reinforcement learning exist?. ICLR 2023.
> [3]: Park, Seohong, Tobias Kreiman, and Sergey Levine. "Foundation policies with hilbert representations." ICML (2024).
> [4]: Loper, Matthew, et al. "SMPL: A skinned multi-person linear model." Seminal Graphics Papers: Pushing the Boundaries, Volume 2. 2023. 851-866.
>
>
> ----------------
>
> Please let us know if there are any further questions and we would be happy to answer them!

---

### Official Review · Reviewer_Eidy · 2025-11-01

**Soundness:** 3
**Presentation:** 3
**Contribution:** 3
**Rating:** 6
**Confidence:** 3

**Summary:**

This paper introduces **RLDP (Regularized Latent Dynamics Prediction)** as a pretraining objective for zero-shot reinforcement learning.
The main contributions include:

1. Proposing RLDP, a method to pretrain state representations for zero-shot RL via latent next-state prediction with regularization.
2. Theoretically proving that training in the latent space can approximate training in the original state space, preserving the predictive power of successor measures.
3. Conducting extensive experiments under both online and offline settings, showing that RLDP achieves competitive performance.
4. Demonstrating the advantage of RLDP under low-coverage conditions, where other methods tend to fail.

**Strengths:**

1. Comprehensive experimental settings, including both online and offline environments.
2. Theoretical guarantees that connect the latent training objective to successor-measure consistency.
3. Rich intuition and ablation studies that help interpret how RLDP improves representation quality.

**Weaknesses:**

There are no major flaws, but several aspects could be improved for clarity and consistency.

1. Structure and exposition could be better organized:
   - The paper does not clearly specify which parameters are optimized for each loss.
     The training process is only briefly introduced around line 295, and it’s unclear which components receive gradients.
     A concise summary of the full optimization pipeline (e.g., which modules update under each loss) would make the method section more readable.
   - In lines 180–190 describing policy improvement, it is unclear whether a separate policy network is maintained.
     In Equation (4), π(s) is written as if it equals a Q-value—perhaps “max” should be “argmax”?
     Equation (5) is understandable but would be more precise if an expectation operator **E[…]** were added.
   - The variable $\bar{M}$ is first defined in *Lemma 4.3* but appears earlier in Equation (3).
     This may indicate an input mismatch: it should likely take $\Phi(s′)$ instead of $s′$ as argument.

2. The claim in line 198—“these representations minimize the prediction error for successor measures for any policy”—is not directly demonstrated.
   Subsequent quantitative analyses focus on returns rather than direct prediction errors.
   Since this is primarily a representation learning method, showing explicit results for prediction errors (e.g., L2 distance between predicted and ground-truth successor measures) would greatly strengthen the argument.

**Questions:**

1. In Figure 1, the paper mentions a “mild form of collapse.”
   However, the cosine similarity range of 0.4–0.6 does not obviously indicate collapse. Why is this value considered high?  Providing reference baselines (e.g., expected cosine similarity for random embeddings) would make this claim more convincing.

2. The paper evaluates RLDP on D4RL’s medium and medium-expert datasets.
   Why not also include medium-replay? Those datasets have broader coverage, which could provide further insight into RLDP’s robustness under high-variance data.

---

> ### Author Response · Authors · 2025-11-24
> **Response to Reviewer Eidy**
>
> We thank the reviewer for their detailed feedback on our work and encouraging comments. We are happy to learn that the reviewer found our method theoretically sound with extensive empirical study as well as beneficial in low-coverage settings.
>
> > The paper does not clearly specify which parameters are optimized for each loss.
> > In lines 180–190 describing policy improvement, it is unclear whether a separate policy network is maintained.
>
> These are great suggestions. We have updated the method section 4 of the paper to clearly show which parameters are optimized for each loss. We hope that our Algorithm 1 in appendix A.6 also adds clarity explaining the complete method.
>
> > “these representations minimize the prediction error for successor measures for any policy”—is not directly demonstrated.Subsequent quantitative analyses focus on returns rather than direct prediction errors.Since this is primarily a representation learning method, showing explicit results for prediction errors (e.g., L2 distance between predicted and ground-truth successor measures) would greatly strengthen the argument.
>
> We did not intend to claim that the representations minimize the prediction error for successor measures for any policy. Lemma 4.3 states that $L_{RLDP}$ upper bounds the prediction error of the successor measure for all policies. Minimizing $L_{RLDP}$ will lead to a reduction in the prediction error but not necessarily will lead to minimum prediction error. We thank the reviewer for pointing this out and modify the claim in the paper.
>
>
> > In Figure 1, the paper mentions a “mild form of collapse.” However, the cosine similarity range of 0.4–0.6 does not obviously indicate collapse. Why is this value considered high? Providing reference baselines (e.g., expected cosine similarity for random embeddings) would make this claim more convincing.
>
> This is a nice idea and we have added Figure 6 which shows cosine similarity with orthogonality regularization. This shows the effect of diversity regularization clearly in preventing mild collapse. We note that cosine similarity for random embeddings are: a) 0 for a embeddings uniformly sampled on the hypersphere $\mathbb{S}^{d-1}$ b) the initial value in the plot for embeddings generated by randomly initialized neural network (since before we start training our NNs are randomly initialized). With respect to both these baselines, increasing feature similarity can be intuitively seen as high
>
> > Writing improvements
>
> We thank the reviewer for suggesting writing improvements and have updated the paper.
>
> >The paper evaluates RLDP on D4RL’s medium and medium-expert datasets.Why not also include medium-replay? Those datasets have broader coverage, which could provide further insight into RLDP’s robustness under high-variance data.
>
> Thanks for the recommendation. We primarily chose medium and medium-expert to represent datasets with narrow coverage. Medium-replay datasets have broader coverage as they contain all transitions from the replay buffer of the agent encountered during its exploration. We have added medium-replay datasets in our comparison (Table 10) and find that RLDP is competitive to baselines for medium-replay datasets.
>
>
> --------------
> Please let us know if there are any further questions and we would be happy to answer them!

---

### Official Review · Reviewer_2m5w · 2025-11-01

**Soundness:** 4
**Presentation:** 3
**Contribution:** 3
**Rating:** 8
**Confidence:** 4

**Summary:**

This paper introduces a novel representation learning objective for unsupervised RL, where rewards are not given in the data, focused on enabling downstream Behavioral Foundation Models to perform effective zero-shot RL. Their objective broadly encompasses two components: a next latent state prediction loss motivated by theory, and an orthogonal regularization loss motivated through empirical observation. Through a series of experiments in robotics simulators, they validate their method, and show that their simple framework is a strong baseline for learning BFMs in a variety of settings including offline zero-shot RL, online zero-shot RL, and low coverage datsets.

**Strengths:**

- Their proposed method is well-motivated both theoretically and empirically, and the authors present strong experimental evidence for their framework relative to other works in its area.
- The experiments in their paper suggests that their proposed method can produce competitive results in the zero-shot RL setting, all the while being both simple to understand and implement compared to existing methods. Thus, this work is an important entry point for new researchers in the field of BFMs for zero-shot RL, and an important reference for existing researchers as well.
- Their method is further studied beyond final performances, as the authors provide ablation studies for almost all components of their framework in either the main text or the supplementary material.

**Weaknesses:**

- It would be nice to further contextualize RLDP with the existing representation learning methods, by either providing the exact objectives of the baselines used in the paper(Laplace, FB, HILP, PSM) or being more explicit about the different assumptions used. For instance, the authors also suggest that their method is simpler in implementation and lacks some assumptions made by prior methods such as a prior class of policies. Showing the objectives used in the other works would clarify these differences

**Questions:**

- Aside from ease of implementation, what other benefits arise from the simplicity of the proposed methods? Can we also any significant improvements in compute costs or sample efficiency?
- As remarked in [1], next latent state prediction is generally a strong objective for representation learning in RL. However, in some tasks, other objectives such as observation reconstruction can also be beneficial. Due to the similar nature of all the environments in the paper (robotics physics-based simulators), it is possible that the proposed objective is only a strong baseline in environments of that class. For example, [1] remarks that while observation reconstruction is not beneficial in MuJoCo tasks, it can be good in low-dimensional clean environments such as grid worlds. Have you considered the benefit of other simple and popular representation learning objectives, and possibly experimenting with other environments such as pixel-based environments?


### Minor Comments
- Typo in line 287: "Lemma 4.33 implies that minimizes".
- Typo in line 370: "Due to **the** exploratory challenge [...]"
- Typo in line 412: "a class of **policies** to learn [...]"
- The text is not consistent with its paragraph titles. For example, paragraph titles are sometimes bolded with a period (e.g. lines 320, 347, 353, 368 ...), bolded with a colon (e.g. lines 385, 423, 431), or italicized (e.g. lines 441 and 448).
- The text is not consistent with line breaks between paragraphs. There looks to be missing line breaks in line 368, 375, 385, 423, 430, 440.
- Figure 3 does not show any variance statistics for the "average return" datapoints. It can be difficult to parse whether the stated improvement from the orthogonality regularization is statistically significant.
- It could be benficial to further contextualize the results by adding one or two naive baselines that are not state-of-the-art. For example, what is the performance of zero-shot RL on a random representations?
- The results in Figure 2 are difficult to parse both in detail and at a glance. It could be beneficial to aggregate these results by normalizing the scores and providing an IQM [2]


### References
[1] Ni et al., 2023 -  https://arxiv.org/pdf/2401.08898
[2] Agarwal et al., 2022 - https://arxiv.org/pdf/2108.13264

---

> ### Author Response · Authors · 2025-11-24
> **Response to Reviewer 2m5w**
>
> We thank the reviewer for their detailed feedback on our work and encouraging comments. We are happy to learn that the reviewer found our method well motivated theoretically and empirically, as well as found our ablations extensive.
>
> > It would be nice to further contextualize RLDP with the existing representation learning methods,
>
> This is a great suggestion, we have added a subsection in the appendix (A.2) with a pointer from preliminary section [line 193] explaining existing methods which will help the readers understand clearly how RLDP fits into the wider literature.
>
> >Aside from ease of implementation, what other benefits arise from the simplicity of the proposed methods? Can we also any significant improvements in compute costs or sample efficiency?
>
>
>
>
>
>
>
>
> RLDP uses slightly less compute, though not by a lot. Two of the baselines we compare with (PSM and FB) use two additional networks in the representation learning phase, which can be more computationally expensive.
>
>
>
>
> >As remarked in [1], next latent state prediction is generally a strong objective for representation learning in RL. However, in some tasks, other objectives such as observation reconstruction can also be beneficial. Due to the similar nature of all the environments in the paper (robotics physics-based simulators), it is possible that the proposed objective is only a strong baseline in environments of that class. For example, [1] remarks that while observation reconstruction is not beneficial in MuJoCo tasks, it can be good in low-dimensional clean environments such as grid worlds. Have you considered the benefit of other simple and popular representation learning objectives, and possibly experimenting with other environments such as pixel-based environments?
>
> Yes, as the reviewer points out most of our domains are physics-based simulators, but we would like to point out that their dynamics can vary significantly. One existing domain in our paper is Pointmass which provides a low-dimensional observation/action space and simple dynamics. We did further qualitative analysis of the representations learned in Figure 7 which shows how the representations look in a projected space.  We add results on how successor measures learned with RLDP looks like on gridworld in appendix A.8. While reconstruction might be a good objective for low-dimensional environments, we highlight that a number of practical settings (eg. humanoid embodiment) considered in this work deal with large observation spaces and reconstructing observations might be overly restrictive. For instance, learning representations for autonomous driving tasks in different lighting conditions (an irrelevant factor) would suffer in generalization if we are to learn representations that also reconstruct the lighting conditions. Prior work’s [1] extensive comparison demonstrates that autoencoding and random feature baselines do not fare well for zero-shot RL in a variety of domains (including Maze which is a 2D domain). We chose DMC (4 environments, 16 tasks), Humanoid (45 tasks), D4RL (3 environment, 6 tasks) with up to 358 dimensional observations and 69 dimensional actions as a diverse set of environments. A detailed study of pixel-based environments would be an interesting direction for future work.
>
> > Minor Comments
>
> We thank the reviewer for suggesting the writing improvements and have updated the paper.
>
> > Figure 3 does not show any variance statistics for the "average return" datapoints.
>
> We have added std-deviations for Figure 3. We do one-sided Mann–Whitney U tests on the per-seed returns to compare different values of the orthogonality regularization, and we observe that orthogonality regularization (coefficient 1.0) has statistically significant improvements in most environments over coefficient=0.
>
> > It could be beneficial to further contextualize the results by adding one or two naive baselines that are not state-of-the-art.
>
> We have added a random feature baseline for zero-shot RL [appendix table 7]. Random features have low performance across most tasks.
>
> > The results in Figure 2 are difficult to parse both in detail and at a glance
>
>  To improve readability of results in Figure 2, we have added a violin plot with IQM [fig. 9] that compares oracle-normalized scores of RLDP w.r.t baseline methods
>
> [1]: Touati, Ahmed, Jérémy Rapin, and Yann Ollivier. "Does zero-shot reinforcement learning exist?." ICLR 2023.
>
> --------------
> Please let us know if there are any further questions and we would be happy to answer them!

---

### Author Response · Authors · 2025-12-03
**General Response**

This work introduces Regularized Latent Dynamics Prediction (RLDP), a simple representation learning method for unsupervised RL based on latent next-state prediction with an orthogonality regularizer. We evaluate RLDP against state-of-the-art Behavioral Foundation Model methods (FB, PSM, Laplace) on a broad suite of benchmarks (16 DMC tasks, 45 Humanoid, and 6 D4RL tasks) with up to 358 dimensional observations and 69 dimensional actions, and find RLDP to be competitive with or better than existing methods.

The reviews broadly agree to the strengths of our paper, and find that RLDP is:

* (2m5w, Eidy, GfJN) **theoretically motivated and sound**, with a clear connection between the latent dynamics objective and successor measures
* (2m5w, Th7h, GfJN) **simple yet effective**, and comparatively easier to understand and implement
* (2m5w, Eidy, Th7h, GfJN) **evaluated thoroughly on a variety of settings** (including online and offline zero-shot RL), and performs well, especially in low-coverage settings where other methods tend to struggle
* (2m5w) **an important reference** for existing researchers and entry point for new researchers in the zero-shot RL community

The concerns raised are primarily about clarity and additional context, and the reviews raise no major objections. They have been addressed as follows:

Writing changes:
* Based on suggestions from Reviewers 2m5w, Eidy, we have made writing improvements to the paper: the method section (section 4) has been updated to clearly indicate the parameters our RLDP objective is optimizing for. We have added a section in the appendix (A.2) explaining existing SOTA methods which will help readers contextualize RLDP w.r.t the wider literature.

Clarification to existing experimental results and figures:
* Based on suggestions from Reviewer 2m5w, we have updated fig 3 in the main paper to include variance statistics, and added fig 9 to display aggregate humanoid results.
* Reviewer Eidy suggested including a reference baseline to indicate why a cosine similarity in range 0.4-0.6 would be considered high and indicate a “mild form of collapse” (ref: section 4.1). We have added fig 6 which includes a reference baseline of RLDP-learned representations that shows low cosine similarity.

Additional baselines and experimental results:
* Following Reviewer 2m5w, we added a naive random-feature zero-shot RL baseline and experiments in low-dimensional gridworld environment. These are in Table 7 and Appendix section A.8 respectively, which has qualitative visualizations of successor measures with RLDP.
* Following Reviewer Eidy, we added results in D4RL medium-replay datasets in Table 10, showing that RLDP remains competitive in broader-coverage datasets.
* Following Reviewer GfJN, we added an experiment where we learn a Universal Successor Feature on top of RLDP state features (Table 8). These results support the applicability of RLDP representations beyond the specific policy-learning methods considered in the main text.

These revisions address the specific questions asked by the reviewers without altering the core RLDP method or empirical conclusions.

We thank the reviewers for their feedback and suggestions.

---

### Meta-Review · Area_Chair_G8E1 · 2026-01-07

**Summary:**

Overall, reviewers agree this is a simple but surprisingly strong baseline for learning BFM-style representations via latent next-state prediction + orthogonality regularization, and that the paper is theoretically motivated and evaluated broadly, including settings where other methods struggle (notably low-coverage).
The main things holding the paper back pre-rebuttal were (i) clarity/organization in the method section (what gets updated by which loss; a couple of notation/claim issues), (ii) contextualization vs prior baselines, and (iii) requests for a few missing “sanity” baselines/aggregations and robustness checks (variance stats, IQM-style aggregation, random features, broader D4RL coverage, etc.).
This reads like a paper that’s useful for the community and mostly got “presentation + completeness” feedback rather than fundamental objections. Given how much of that was addressed in the rebuttal, my suggested decision is Accept (poster).

**Reviewer Concerns:**

Addressed (or largely addressed) by the rebuttal:
-	Better contextualization of baselines/assumptions: authors added an appendix subsection explaining existing SOTA methods and how RLDP fits.
-	Clarity on optimization / what receives gradients: method section updated to clearly indicate which parameters each loss optimizes; algorithmic clarification added.
-	Variance/readability of main plots: Fig 3 updated with variance stats; aggregate humanoid results + IQM/violin-style summary added to improve readability.
-	Naive baseline request (random features): added random-feature zero-shot RL baseline; random features perform poorly.
-	“Mild collapse” / cosine similarity interpretation: added a reference baseline figure and discussed what cosine similarity looks like for random/untrained embeddings.
-	Broader D4RL coverage (medium-replay): added medium-replay results and report competitive performance there.
-	Theory–practice mismatch + broader applicability (GfJN): authors directly address the forward/backward feature issue and added an experiment learning USF on top of RLDP features.
Still outstanding/not fully resolved:
-	Direct evaluation of successor-measure prediction error: Reviewer Eidy explicitly asked for prediction-error style evidence (not just return), and the rebuttal mostly reframes the claim (upper bound vs “minimize”) rather than adding that quantitative evidence.
-	Generalization beyond the current benchmark class: concerns remain that evaluation is still primarily simulated continuous control; “pixel-based” or other qualitatively different domains are left as future work.
-	Real-world/transfer: Th7h notes lack of real-world evaluation; authors provide reasonable discussion, but it’s still discussion.

**Reviewer Scores:**

-	Reviewer 2m5w (score: 8 accept): likely stays 8. Essentially all of their “make it clearer/add sanity baselines/add variance + IQM” requests were addressed.
-	Reviewer Eidy (score: 6 marginally above the acceptance threshold): likely stays 6. Their clarity/claim concerns were addressed (method clarity + claim softened), and the added baselines/medium-replay directly respond. The only remaining gap is the missing “prediction error” style measurement, which I think would keep them from raising score.
-	Reviewer Th7h (score: 6 marginally above the acceptance threshold): likely stays 6. The rebuttal is fine, but it doesn’t actually add real-world evidence.
-	Reviewer GfJN (score: 6 marginally above the acceptance threshold): likely 6 to 8. The rebuttal directly addresses their theory/practice mismatch concern and runs the USF-on-top experiment they asked for, which is exactly the kind of thing that usually shifts a borderline review upward.

---

### Decision · Program_Chairs · 2026-01-26

Accept (Poster)